# A replica analysis of under-bagging

**Takashi Takahashi**[*]
**Institute for Physics of Intelligence,**
**The University of Tokyo**

**takashi-takahashi@g.ecc.u-tokyo.ac.jp**

Reviewed on OpenReview: **https://openreview.net/forum?id=7HIOUZAoq5**

## Abstract

Under-bagging (UB), which combines under-sampling and bagging, is a popular ensemble learning method for training classifiers on an imbalanced data. Using bagging to reduce the increased variance caused by the reduction in sample size due to under-sampling is a natural approach. However, it has recently been pointed out that in generalized linear models, naive bagging, which does not consider the class imbalance structure, and ridge regularization can produce the same results. Therefore, it is not obvious whether it is better to use UB, which requires an increased computational cost proportional to the number of under-sampled data sets, when training linear models. Given such a situation, in this study, we heuristically derive a sharp asymptotics of UB and use it to compare with several other popular methods for learning from imbalanced data, in the scenario where a linear classifier is trained from a two-component mixture data. The methods compared include the under-sampling (US) method, which trains a model using a single realization of the under-sampled data, and the simple weighting (SW) method, which trains a model with a weighted loss on the entire data. It is shown that the performance of UB is improved by increasing the size of the majority class while keeping the size of the minority fixed, even though the class imbalance can be large, especially when the size of the minority class is small. This is in contrast to US, whose performance is almost independent of the majority class size. In this sense, bagging and simple regularization differ as methods to reduce the variance increased by under-sampling. On the other hand, the performance of SW with the optimal weighting coefficients is almost equal to UB, indicating that the combination of reweighting and regularization may be similar to UB.

## 1 Introduction

In the context of binary classification, class imbalance refers to the situations where one class overwhelms the other in terms of the number of data points. Learning on class imbalanced data often occurs in real-world classification tasks, such as medical data analysis (Cohen et al., 2006), image classification (Khan et al., 2018), and so on. The primary goal when training classifiers on such imbalanced data is to achieve good generalization to both minority and majority classes. However, standard supervised learning methods without any special care often results in poor generalization performance for the minority class. Therefore, special treatments are necessary and have been one of the major research topics in the machine learning (ML) community for decades.

Under-bagging (UB) (Wallace et al., 2011) is a popular and efficient method for dealing with a class imbalance that combines under-sampling (US) and bagging (Breiman, 1996). The basic idea of UB is to deal with class imbalance using under-sampling that randomly discards a portion of the majority class data to ensure an

---

[*]takashi-takahashi@g.ecc.u-tokyo.ac.jp

equal number of data points between the minority and the majority classes. However, this data reduction increases the variance of the estimator, because the total number of the training examples is reduced. To mitigate this, UB aggregates estimators for different under-sampled data sets, similar to the aggregation performed in bootstrap aggregation (Breiman, 1996). US corresponds to a method of training a classifier on a single undersampled data. (Wallace et al., 2011) argues that UB is particularly effective for training of overparameterized models, where the number of the model parameters far exceeds the size of the training data.

There is another line of approach, that is, cost-sensitive methods, which modify the training loss at each data point depending on the class it belongs to. However, it has recently been argued that training overparameterized models using cost-sensitive methods requires delicate design of loss functions (Lin et al., 2017; Khan et al., 2018; Byrd & Lipton, 2019; Ye et al., 2020; Kini et al., 2021) instead of the naive weighted cross-entropy (King & Zeng, 2001) and so on. In contrast to such delicate cost-sensitive approaches, UB has the advantage of being relatively straightforward to use, since it can achieve complete class balance in any under-sampled dataset.

Bagging is a natural approach to reduce the increased variance of weak learners, resulting from the smaller data size due to the resampling. However, unlike decision trees, which are the primary models used in bagging, neural networks can use simple regularization methods such as the ridge regularization to reduce the variance. In fact, it has recently been reported experimentally that naive bagging, which ignores the structure of the class balance, does not improve the classification performance of deep neural networks (Nixon et al., 2020). Furthermore, in generalized linear models (GLMs), it has been reported both theoretically (Sollich & Krogh, 1995; Krogh & Sollich, 1997; LeJeune et al., 2020; Du et al., 2023) and experimentally (Yao et al., 2021) that bagging with a specific sampling rate is exactly equivalent to the ridge regularization with a properly adjusted regularization parameter. These results indicate that naive bagging, which ignores the class imbalance structure, plays the role of an implicit ridge regularization in linear models. Given these results, the question arises whether it is better to use UB, which requires an increased computational cost proportional to the number of under-sampled data sets, when training linear models or neural networks, rather than just using US and the ridge regularization.

Given the above situation, this study aims to clarify whether bagging or ridge regularization should be used when learning a linear classifier from a class-imbalanced data. Specifically, we compare the performance of classifiers obtained by UB with those obtained by (i) the under-sampling (US) method, which trains a model using a single realization of the subsampled dataset, and (ii) the simple weighting (SW) method, which trains a model with a weighted loss on the entire data, when training a linear model on two-component mixture data, through the $F$-measure, the weighted harmonic mean of specificity and recall (see Section 2 for the formal definition). To this end, we heuristically derive a sharp characterization of the classifiers obtained by minimizing the randomly reweighted empirical risk in the asymptotic limit where the input dimension and the parameter dimension diverge at the same rate, and then use it to analyze the performance of UB, US, and SW.

The contributions of this study are summarized as follows:

- A sharp characterization of the statistical behavior of the linear classifier is derived, where the classifier is obtained by minimizing the reweighted empirical risk function on two-component mixture data, in the asymptotic limit where input dimension and data size diverge proportionally (Claim 3). This includes the properties of the estimator obtained from the under-sampled data and the simple weighted loss function. There, the statistical property of the logits for a new input is described by a small finite set of scalar quantities that is determined as a solution of nonlinear equations, which we call *self-consistent equations* (Definition 1). The sharp characterization of the $F$-measure is also obtained (Claim 4). The derivation is based on the replica method of statistical physics (Mézard et al., 1987; Charbonneau et al., 2023; Montanari & Sen, 2024).

- Using the derived sharp asymptotics, it is shown that the estimators obtained by UB have higher generalization performance in terms of $F$-measure than those obtained by US. Although the performance of UB in terms of the $F$-measure is improved by increasing the size of the majority class while

keeping the size of the minority class fixed, the performance of US is almost independent of the size of the excess examples of the majority class (Section 4.1 and 4.3). This result is clearly different from the case of the naive bagging in training GLMs without considering the structure of the class imbalance, where the ensembling and ridge regularization yield the same performance. That is, the ensembling and the regularization are different when using under-sampled data.

- On the other hand, the performance of SW with carefully optimized weighting coefficients is almost equal to UB (Section 4.2). In this sense, the combination of the weighting and ridge regularization is similar to UB. However, in SW, if the weighting coefficients are not well optimized and the default values in ML packages such the scikit-learn are used (Pedregosa et al., 2011), the performance deteriorates rapidly as the number of exess majority samples increases (Section 4.2 and Appendix D), although UB does not requires such careful tuning.

- Furthermore, it is shown that UB is robust to the existence of the interpolation phase transition, where data goes from linearly separable to inseparable training data (Section 4.1.1). In contrast, the performance of the interpolator obtained from a single realization of balanced data is affected by such a phase transition as reported in the previous study (Mignacco et al., 2020).

## 1.1 Further related works

Learning under class imbalance is a classic and notorious problem in data mining and ML. In fact, the development of dealing with imbalanced data was selected as "10 CHALLENGING PROBLEMS IN DATA MINING RESEARCH" (Yang & Wu, 2006) about 20 years ago in the data mining community. Due to its practical relevance, a lot of methodological research has been continuously conducted and are summarized in reviews (He & Garcia, 2009; Longadge & Dongre, 2013; Johnson & Khoshgoftaar, 2019; Mohammed et al., 2020).

Learning methods for imbalanced data can broadly be classified into two categories: cost- and data-level ones. In cost-sensitive methods, the cost function itself is modified to assign different costs to each class. A classic method includes a weighting method that assigns a different constant multiplier to the loss at each data point depending on the class it belongs to. However, when training overparameterized models, it has been argued that a classic weighting method fails and a more delicate design of the loss function is required to properly adjust the margins between classes (Lin et al., 2017; Khan et al., 2018; Byrd & Lipton, 2019; Ye et al., 2020; Kini et al., 2021). In data-level methods, the training data itself is modified to create a class-balanced dataset using resampling methods. The major approaches include US (Drummond et al., 2003), which randomly discards some of the majority class data, and SMOTE (Chawla et al., 2002), which creates synthetic minority data points. Although the literature (Wallace et al., 2011) argues that a combination of US and bagging is more effective for learning overparameterized models than the naive SW method, the comparison with other regularization techniques is not discussed. Intuitively, when the number of data points is small and the data is linearly separable, the position of the classification plane (i.e., the bias) cannot be uniquely determined without regularization even if the cost function is weighted by a multiplicative factor. In this situation, resampling can be more efficient to control the bias term than the simple weight decay. On the other hand, this intuitive picture cannot explain how much performance difference would be observed quantitatively until the values are evaluated precisely. Showing this difference qualitatively is one goal of this paper.

Technically, our work is an analysis of the sharp asymptotics of resampling methods in which the input dimension and the size of the data set diverge at the same rate. The analysis of ML methods in this asymptotic regime has been actively carried out since the 1990s, using techniques from statistical physics (Opper & Kinzel, 1996; Opper, 2001; Engel & Van den Broeck, 2001). Recently, it has become one of the standard analytical approaches, and various mathematical techniques have also been developed to deal with it. The salient feature of this proportional asymptotic regime is that the macroscopic properties of the learning results, such as the predictive distribution and the generalization error, do not depend on the details of the realization of the training data[1], when using convex loss. That is, the fluctuations of

---

[1] In contrast, microscopic quantities such as each element of the weight vector fluctuate depending on each realization of the training data.

these quantities with respect to the training data vanish, allowing us to make sharp theoretical predictions (Barbier et al., 2019; Charbonneau et al., 2023). The technical tools for analyzing such asymptotics include the replica method (Gerace et al., 2020; Charbonneau et al., 2023; Montanari & Sen, 2024), convex Gaussian minimax theorem (Thrampoulidis et al., 2018), random matrix theory (Mei & Montanari, 2022), adaptive interpolation method (Barbier & Macris, 2019; Barbier et al., 2019), approximate message passing (Sur & Candès, 2019; Feng et al., 2022).

Classification of Gaussian mixture data using linear models has been extensively studied in such a proportional asymptotic regime (Mignacco et al., 2020; Loureiro et al., 2021; Mannelli et al., 2024). (Mignacco et al., 2020) studies empirical risk minimization on binary Gaussian mixture models and reports that ridge regularization with an infinitely large regularization parameter yields the Bayes optimal classifier for balanced clusters with the same covariance structure. (Loureiro et al., 2021) extends the results to $K$-component Gaussian mixture models with $K > 2$ and investigates the role of regularization in more detail. (Pesce et al., 2023) investigates when and how the phenomenology found in the above solvable setups is shared with real-world setups. Also, (Mannelli et al., 2024) investigates a classification problem when the ground truth labels are given by cluster dependent teacher models especially from the viewpoint of fairness. Our work can be seen as an extension of these analyses to ensemble learning in the simplest setup.

Resampling methods have also been studied in the same proportional asymptotics. (Sollich & Krogh, 1995; Krogh & Sollich, 1997; Malzahn & Opper, 2001; 2002; 2003; LeJeune et al., 2020; Ando & Komaki, 2023; Bellec & Koriyama, 2024; Clarté et al., 2024; Patil & LeJeune, 2024) analyze the performance of the ensemble average of the linear models, and (Takahashi, 2023) analyzes the distribution of the ensemble average of the de-biased elastic-net estimator (Javanmard & Montanari, 2014a;b). However, these analyses do not address the structure of the class imbalance. (Loffredo et al., 2024) investigates the under/over sampling for linear support vector machines and reports that the combination of under/over sampling yields the optimal result at severely label imbalanced cases, although the effect of ensembling is not considered. Ensembling in random feature models as a method to reduce the effect of random initialization are also considered in (D'Ascoli et al., 2020; Loureiro et al., 2022). Furthermore, the literature (Obuchi & Kabashima, 2019; Takahashi & Kabashima, 2019; 2020) derives the properties of the estimators trained on the resampled data set in the analysis of the dynamics of approximate message passing algorithms that yield exact results in Gaussian or rotation-invariant feature data.

## 1.2 Notations

Throughout the paper, we use some shorthand notations for convenience. We summarize them in Table 1.

## 2 Problem setup

This section presents the problem formulation and our interest in this work. First, the assumptions about the data generation process are described, and then the estimator of interest is introduced.

Let $D^+ = \{(\boldsymbol{x}_\mu^+, 1)\}_{\mu=1}^{M^+}, \boldsymbol{x}_\mu^+ \in \mathbb{R}^N$ and $D^- = \{(\boldsymbol{x}_\nu^-, -1)\}_{\nu=1}^{M^-}, \boldsymbol{x}_\nu^- \in \mathbb{R}^N$ be the sets of positive and negative examples; in total we have $M = M^+ + M^-$ examples. In this study, the positive class is assumed to be the minority class, that is, $M^+ < M^-$. Also, it is assumed that positive and negative samples are drawn from a mixture model whose centroids are located at $\pm \boldsymbol{v}/\sqrt{N}$ with $\boldsymbol{v} \in \mathbb{R}^N$ being a fixed vector as follows:

$$\boldsymbol{x}_\mu^+ = \frac{1}{\sqrt{N}}\boldsymbol{v} + \boldsymbol{z}_\mu^+, \tag{1}$$

$$\boldsymbol{x}_\nu^- = -\frac{1}{\sqrt{N}}\boldsymbol{v} + \boldsymbol{z}_\nu^-, \tag{2}$$

where $\boldsymbol{z}_\mu^+$ and $\boldsymbol{z}_\nu^-$ are noise vectors. We assume that $z_{\mu,i}^+, z_{\nu,i}^- \sim_{\text{iid}} p_z, \mu \in [M^+], \nu \in [M^-], i \in [N]$, where the mean and the variance of $p_z$ are zero and $\Delta$, respectively, and also the higher order moments are finite. From the rotational symmetry, we can fix the direction of the vector $\boldsymbol{v}$ as $\boldsymbol{v} = (1, 1, \ldots, 1)^\top$ without loss of generality. The goal is to obtain a classifier that generalize well to both positive and negative inputs from $D = D^+ \cup D^-$. The performance metric for classifiers cannot be uniquely determined because they

| Notation | Description |
|----------|-------------|
| $N$ | input dimension |
| $M^+$ | size of the positive examples (assumed to be the minority class) |
| $M^-$ | size of the negative examples (assumed to be the majority class) |
| $M$ | $M^+ + M^-$, the size of the overall training data |
| $\mu, \nu$ | indices of data points |
| $i, j$ | indices of the weight vector $\boldsymbol{w}$ |
| $[n]$ | for an positive integer $n$, the set $\{1, \ldots, n\}$ |
| $v_i$ | $i$-th element of a vector $\boldsymbol{v}$ |
| $\boldsymbol{x} \cdot \boldsymbol{y}$ | for vectors $\boldsymbol{x}, \boldsymbol{y} \in \mathbb{R}^N$, the inner product of them: $\boldsymbol{x} \cdot \boldsymbol{v} = \sum_{i=1}^{N} x_i y_i$. |
| $\boldsymbol{1}_N$ | an $N$-dimensional vector $(1, 1, \ldots, 1) \in \mathbb{R}^N$ |
| $\mathbb{1}(\cdot)$ | indicator function |
| $\mathcal{N}(\mu, \sigma^2)$ | Gaussian density with mean $\mu$ and variance $\sigma^2$ |
| $\text{Poisson}(\mu_*)$ | Poisson distribution with mean $\mu_*$ |
| iid | independent and identically distributed |
| $\mathbb{E}_{X \sim p_X}[f(X)]$ | Expectation regarding random variable $X$ |
| | where $p$ is the density function for the random variable $X$ |
| | (lower subscript $X \sim p_X$ or $p_X$ can be omitted if there is no risk of confusion) |
| $\mathbb{V}_{X \sim p_X}[f(X)]$ | Variance: $\mathbb{E}[f(X)^2] - \mathbb{E}[f(X)]^2$ |
| | (lower subscript $X \sim p_X$ or $p_X$ can be omitted if there is no risk of confusion) |
| $\delta_\text{d}(\cdot)$ | Dirac's delta function |

Table 1: Notations

can depend on factors such as how much importance is given to errors for each class. For simplicity, in this work, we take the position that errors are treated equally for positive and negative examples, and measure performance by the $F$-measure, which is defined as the harmonic mean of specificity $\mathcal{S}$ and recall $\mathcal{R}^2$. Specificity and recall are the expected error rates conditioned that inputs are positive and negative examples, respectively. Let $\hat{y}(\boldsymbol{x})$ be the predicted label for an input $\boldsymbol{x}$. Then, specificity and recall are defined as

$$\mathcal{S} = \mathbb{E}_{\boldsymbol{z} \sim p_z}[\mathbb{1}(\hat{y}(\boldsymbol{x}^+) \neq 1)], \quad \boldsymbol{x}^+ = \frac{1}{\sqrt{N}} \boldsymbol{v} + \boldsymbol{z}, \tag{3}$$

$$\mathcal{R} = \mathbb{E}_{\boldsymbol{z} \sim p_z}[\mathbb{1}(\hat{y}(\boldsymbol{x}^-) \neq -1)], \quad \boldsymbol{x}^- = -\frac{1}{\sqrt{N}} \boldsymbol{v} + \boldsymbol{z}, \tag{4}$$

Using these two quantities, the $F$-measure $\mathcal{F}$ is given as

$$\mathcal{F} = \frac{2}{\frac{1}{\mathcal{S}} + \frac{1}{\mathcal{R}}}. \tag{5}$$

This is independent of the label imbalance of the target distribution since specificity and recall are defined thorough the conditional distributions.

We focus on the training of the linear model, i.e., the model's output $f(\boldsymbol{x})$ at an input $\boldsymbol{x}$ is a function of a linear combination of the weight $\boldsymbol{w}$ and the bias $B$:

$$f_\text{model}(\boldsymbol{x}) = f\left(\frac{1}{\sqrt{N}} \boldsymbol{x} \cdot \boldsymbol{w} + B\right), \tag{6}$$

where the factor $1/\sqrt{N}$ is introduced to ensure that the logit $\boldsymbol{x} \cdot \boldsymbol{w}/\sqrt{N} + B$ should be of $\mathcal{O}(1)$, and $f$ is a nonlinear function. In the following, we denote $\boldsymbol{\theta}$ for the shorthand notation of the linear model's parameter.

---

[2]We consider specificity instead of precision here to avoid making the $F$-measure dependent on prevalence.

To obtain a classifier, we consider minimizing the following randomly reweighted empirical risk function:

$$\hat{\boldsymbol{\theta}}(\{c_\mu^+\}, \{c_\nu^-\}, D) = \underset{(\boldsymbol{w}, b) \in \mathbb{R}^{N+1}}{\arg\min} \sum_{\mu=1}^{M^+} c_\mu^+ l^+ (\boldsymbol{x}_\mu^+ \cdot \boldsymbol{w}/\sqrt{N} + b)$$

$$+ \sum_{\nu=1}^{M^-} c_\nu^- l^- (\boldsymbol{x}_\nu^- \cdot \boldsymbol{w}/\sqrt{N} + b) + \lambda \sum_{i=1}^{N} r(w_i), \tag{7}$$

when the bias $B$ is estimated by minimizing the cost function. When the bias is fixed as $\bar{B}$ during the training, one can simply replace $b$ with $\bar{B}$ in (7) and optimize only with respect to $\boldsymbol{w} \in \mathbb{R}^N$. Here, $l^+$ and $l^-$ are the convex loss for positive and negative examples, respectively. For example, the choice of $l^+(x) = -\log(\sigma(x))$ and $l^-(x) = -\log(1 - \sigma(x))$, with $\sigma(\cdot)$ being the sigmoid function, yields the cross-entropy loss. Also, $c_\mu^+ \sim_{\text{iid}} p_c^+$ and $c_\nu^- \sim_{\text{iid}} p_c^-$ are the random coefficients that represent the effect of the resampling or the weighting. In the following, we use the notation $\boldsymbol{c} \equiv \{c_\mu^+\}_{\mu=1}^{M^+} \cup \{c_\nu^-\}_{\nu=1}^{M^-} \sim p_c$. Finally, $\lambda r(x) = \lambda x^2 / 2$ is the ridge regularization for the weight vector with a regularization parameter $\lambda \in (0, \infty)$.

By appropriately specifying the distribution $p_c^\pm$, the estimator (7) can represent estimators for different types of costs for resampled or reweighted data as follows:

- US without replacement (subsampling):

$$p_c^+(c^+) = \delta_{\text{d}}(c^+ - 1) \tag{8}$$

$$p_c^-(c^-) = \mu_* \delta_{\text{d}}(c^- - 1) + (1 - \mu_*)\delta_{\text{d}}(c^-), \tag{9}$$

  where $\mu_* \in (0, 1]$ be the resampling rate. $\mu_* = M^+/M^-$ is a natural guess to balance the size of the sizes of positive and negative examples.

- US with replacement (bootstrap):

$$p_c^+(c^+) = \delta_{\text{d}}(c^+ - 1), \tag{10}$$

$$p_c^- = \text{Poisson}(\mu_*), \tag{11}$$

  where $\mu_* \in (0, 1]$ be the resampling rate.

- Simple weighting:

$$p_c^+(c^+) = \delta_{\text{d}}(c^+ - \gamma^+), \tag{12}$$

$$p_c^-(c^-) = \delta_{\text{d}}(c^- - \gamma^-), \tag{13}$$

  where $\gamma^+, \gamma^- \in (0, \infty)$ is the reweighting coefficient for the positive and negative classes. $\gamma^+ = (M^+ + M^-)/(2M^+), \gamma^- = (M^+ + M^-)/(2M^-)$ is a naive candidate used as a default value in major ML packages such as the scikit-learn (Pedregosa et al., 2011; King & Zeng, 2001).

The ensemble average of the prediction for a new input $\boldsymbol{x}$ is thus obtained as follows:

$$\begin{cases} \mathbb{E}_{\boldsymbol{c}} \left[ \boldsymbol{x} \cdot \hat{\boldsymbol{w}}(\boldsymbol{c}, D)/\sqrt{N} + \hat{B}(\boldsymbol{c}, D) \right], & \text{when bias is estimated} \\ \mathbb{E}_{\boldsymbol{c}} \left[ \boldsymbol{x} \cdot \hat{\boldsymbol{w}}(\boldsymbol{c}, D)/\sqrt{N} + \bar{B} \right], & \text{when bias is fixed} \end{cases} \tag{14}$$

The predicted labels are given by the sign of these averaged quantities. In this study, we mainly focus on the case of subsampling as a method of resampling (8)-(9) except for Section 4.3, which briefly compares the sampling with and without replacement, since the the computational cost is usually smaller than the bootstrap.

The purpose of this study is twofold: the first is to characterize how the prediction $\boldsymbol{x} \cdot \hat{\boldsymbol{w}}(\boldsymbol{c}, D)/\sqrt{N} + \hat{B}(\boldsymbol{c}, D)$ (or $\bar{B}$) depends statistically on $\boldsymbol{x}$ and $\boldsymbol{c}$, and the other is to clarify the difference in performance

with the UB obtained by (14) and the US or SW obtained by (7) with a fixed **c**. We are interested in the generalization performance for both positive and negative inputs. Therefore, we will evaluate the generalization performance using the $F$-measure, the weighted harmonic mean of specificity and recall.

To characterize the properties of the estimators (7), we consider the large system limit where $N, M^+, M^- \to \infty$, keeping their ratios as $(M^+/N, M^-/N) = (\alpha^+, \alpha^-) \in (0, \infty)^2$. In this asymptotic limit, the behavior of the estimators (7) can be sharply characterized as shown in the next section. We call this asymptotic limit as *large system limit* (LSL). In the following, $N \to \infty$ represents the LSL as a shorthand notation to avoid cumbersome notation.

## 3 Asymptotics of the estimator for the randomized cost function

In this section, we present the asymptotic properties of the estimators (7) for the randomized cost functions. It is described by a small finite set of scalar quantities determined as a solution of nonlinear equations, which we refer to as self-consistent equations. The derivation is outlined in Appendix A. Readers unfamiliar with replica analysis may have difficulty in interpreting Definition 1 and Claim 1. In that case, we recommend to proceed to Claims 2-4 first, and then go back to Definition 1 and Claim 1 to understand how the constants are defined.

First, let us define the self-consistent equations that determines a set of scalar quantities $B \in \mathbb{R}, \Theta = (q, m, \chi, v) \in (0, \infty) \times \mathbb{R} \times (0, \infty) \times (0, \infty)$ and $\hat{\Theta} = (\hat{Q}, \hat{m}, \hat{\chi}, \hat{v}) \in (0, \infty) \times \mathbb{R} \times (0, \infty) \times (0, \infty)$ that characterize the statistical behavior of (7).

**Definition 1 (Self-consistent equations)** *The set of quantities $\Theta$ and $\hat{\Theta}$ are defined as the solution of the following set of non-linear equations that are referred to as self-consistent equations. Let $\hat{w}$ and $\hat{u}^\pm$ be the solution of the following one-dimensional randomized optimization problems:*

$$\hat{w} = \arg\min_{w \in \mathbb{R}} \frac{\hat{Q}}{2} w^2 - h_w w + \lambda r(w), \tag{15}$$

$$\hat{u}^\pm = \arg\min_{u \in \mathbb{R}} \frac{u^2}{2\chi\Delta} + c^\pm l^\pm(u + h_u^\pm), \tag{16}$$

*where*

$$h_w = \hat{m} + \sqrt{\hat{\chi}}\xi_w + \sqrt{\hat{v}}\eta_w, \quad \xi_w, \eta_w \sim \mathcal{N}(0, 1), \tag{17}$$

$$h_u^\pm = B \pm m + \sqrt{q}\xi_u + \sqrt{v}\eta_u, \quad \xi_u, \eta_u \sim \mathcal{N}(0, \Delta), \tag{18}$$

*and $c^+ \sim p_c^+, c^- \sim p_c^-$. Then, the self-consistent equations are given as follows:*

$$\chi = \mathbb{E}_{\xi_w, \eta_w \sim \mathcal{N}(0,1)} \left[ \frac{d}{dh_w} \hat{w} \right], \tag{19}$$

$$q = \mathbb{E}_{\xi_w \sim \mathcal{N}(0,1)} \left[ \mathbb{E}_{\eta_w \sim \mathcal{N}(0,1)}[\hat{w}]^2 \right], \tag{20}$$

$$m = \mathbb{E}_{\xi_w, \eta_w \sim \mathcal{N}(0,1)} \left[ \hat{w} \right], \tag{21}$$

$$v = \mathbb{E}_{\xi_w \sim \mathcal{N}(0,1)} \left[ \mathbb{V}_{\eta_w \sim \mathcal{N}(0,1)}[\hat{w}] \right], \tag{22}$$

$$\hat{Q} = -\frac{1}{\chi} \mathbb{E}_{\xi_u, \eta_u \sim \mathcal{N}(0,\Delta), c^\pm \sim p_c^\pm} \left[ \alpha^+ \frac{d}{dh_u^+} \hat{u}^+ + \alpha^- \frac{d}{dh_u^-} \hat{u}^- \right], \tag{23}$$

$$\hat{\chi} = \frac{1}{\Delta\chi^2} \mathbb{E}_{\xi_u \sim \mathcal{N}(0,\Delta)} \left[ \alpha^+ \mathbb{E}_{\eta_u \sim \mathcal{N}(0,\Delta), c^+ \sim p_c^+}[\hat{u}^+]^2 + \alpha^- \mathbb{E}_{\eta_u \sim \mathcal{N}(0,\Delta), c^- \sim p_c^-}[\hat{u}^-]^2 \right], \tag{24}$$

$$\hat{m} = \frac{1}{\Delta\chi} \mathbb{E}_{\xi_u, \eta_u \sim \mathcal{N}(0,\Delta), c^\pm \sim p_c^\pm} \left[ \alpha^+ \hat{u}^+ - \alpha^- \hat{u}^- \right], \tag{25}$$

$$\hat{v} = \frac{1}{\Delta\chi^2} \mathbb{E}_{\xi_u \sim \mathcal{N}(0,\Delta)} \left[ \alpha^+ \mathbb{V}_{\eta_u \sim \mathcal{N}(0,\Delta), c^+ \sim p_c^+}[\hat{u}^+] + \alpha^- \mathbb{V}_{\eta_u \sim \mathcal{N}(0,\Delta), c^- \sim p_c^-}[\hat{u}^-] \right]. \tag{26}$$

*Furthermore, depending on whether the bias is estimated or fixed, the following equation is added to the self-consistent equation*

$$\begin{cases} 0 = \mathbb{E}_{\xi_u, \eta_u \sim \mathcal{N}(0,\Delta), c^\pm \sim p_c^\pm} \left[ \alpha^+ \hat{\mathbf{u}}^+ + \alpha^- \hat{\mathbf{u}}^- \right], & \text{when bias is estimated} \\ B = \bar{B}, & \text{when bias is fixed} \end{cases} \tag{27}$$

It is also noteworthy to comment on the interpretation of the parameter $B$ and $(q, m, v)$ in the self-consistent equations.

**Claim 1 (Interpretation of the parameters $\Theta$)** *At the LSL, the solution of the self-consistent equations are related with the weight vector $\hat{\mathbf{w}}(\mathbf{c}, D)$ as follows:*

$$q = \lim_{N \to \infty} \frac{1}{N} \sum_{i=1}^{N} \mathbb{E}_{\mathbf{c}} \left[ \hat{w}_i(\mathbf{c}, D) \right]^2, \tag{28}$$

$$m = \lim_{N \to \infty} \frac{1}{N} \sum_{i=1}^{N} \mathbb{E}_{\mathbf{c}} \left[ \hat{w}_i(\mathbf{c}, D) \right], \tag{29}$$

$$v = \lim_{N \to \infty} \frac{1}{N} \sum_{i=1}^{N} \mathbb{V}_{\mathbf{c}} \left[ \hat{w}_i(\mathbf{c}, D) \right], \tag{30}$$

$$B = \begin{cases} \lim_{N \to \infty} \hat{B}(\mathbf{c}, D), & \text{when bias is estimated} \\ \bar{B}, & \text{when bias is fixed} \end{cases} \tag{31}$$

*That is, $q, m, v, B$ represents the squared norm of the averaged weight vector, the inner product with the cluster center, the variance with respect to the resampling, and the bias, respectively, at LSL.*

Using the solution of the self-consistent equations, we can obtain the following expression for the average of the prediction.

**Claim 2 ($l$-th moments of the predictor)** *Let $(B, m, q, v)$ be the solution of the self-consistent equation in Definition 1, and let $l = 1, 2, \cdots \in \mathbb{N}$ be the non-negative integer. Also, $\hat{s}^\pm(\mathbf{c}, D) \equiv \mathbf{x}^\pm \cdot \hat{\mathbf{w}}(\mathbf{c}, D)/\sqrt{N} + \hat{B}(\mathbf{c}, D)$ and $\hat{s}^\pm(\mathbf{c}, D) \equiv \mathbf{x}^\pm \cdot \hat{\mathbf{w}}(\mathbf{c}, D)/\sqrt{N} + \bar{B}$ be the predictions of the estimator (7) for an input $\mathbf{x}^\pm$ which is generated as*

$$\mathbf{x}^\pm = \pm \mathbf{1}_N/\sqrt{N} + \mathbf{z}, \quad z_i \sim_{\text{iid}} p_z, i \in [N]. \tag{32}$$

*Then, for any $l$, we can obtain the following result regardless of whether the bias is estimated or not:*

$$\mathbb{E}_{D, \mathbf{x}^\pm} \left[ \mathbb{E}_{\mathbf{c}}[\psi(\hat{s}^\pm)]^l \right] = \mathbb{E}_{\xi_u \sim \mathcal{N}(0,\Delta)} \left[ \mathbb{E}_{\eta_u \sim \mathcal{N}(0,\Delta)}[\psi(B \pm m + \sqrt{q}\xi_u + \sqrt{v}\eta_u)]^l \right], \tag{33}$$

*where $\psi$ is arbitrary as long as the above integrals are convergent.*

This result indicates that the fluctuation with respect to resampling conditioned by $\mathbf{x}$ is effectively described by the Gaussian random variable $\eta_u$, and the fluctuation with respect to the noise term in $\mathbf{x}$ is described by the Gaussian random variable $\xi_u$. Schematically, this behavior can be represented as follows:

$$\hat{s}^\pm = \underbrace{B}_{\text{bias}} + \underbrace{(\pm m)}_{\substack{\text{correlation with} \\ \text{the cluster center}}} + \underbrace{\sqrt{q}\xi_u}_{\substack{\text{fluctuation from} \\ \text{the noise } \mathbf{z}}} + \underbrace{\sqrt{v}\eta_u}_{\substack{\text{fluctuation from} \\ \text{random reweighting } \mathbf{c} \\ \text{conditioned by } \mathbf{x}}}. \tag{34}$$

Therefore, the distribution of the prediction averaged over $K$-realization of the reweighing coefficient $\{\mathbf{c}_k\}_{k=1}^{K}$ can be described as follows.

**Claim 3 (Distribution of the averaged prediction)** *Let $(B, m, q, v)$ be the solution of the self-consistent equation in Definition 1. Then, the distribution of $\hat{s}_K^\pm = \frac{1}{K} \sum_{k=1}^{K} \mathbf{x} \cdot \hat{\mathbf{w}}(\mathbf{c}_k, D)/\sqrt{N} + \hat{B}(\mathbf{c}_k, D)$ (or*

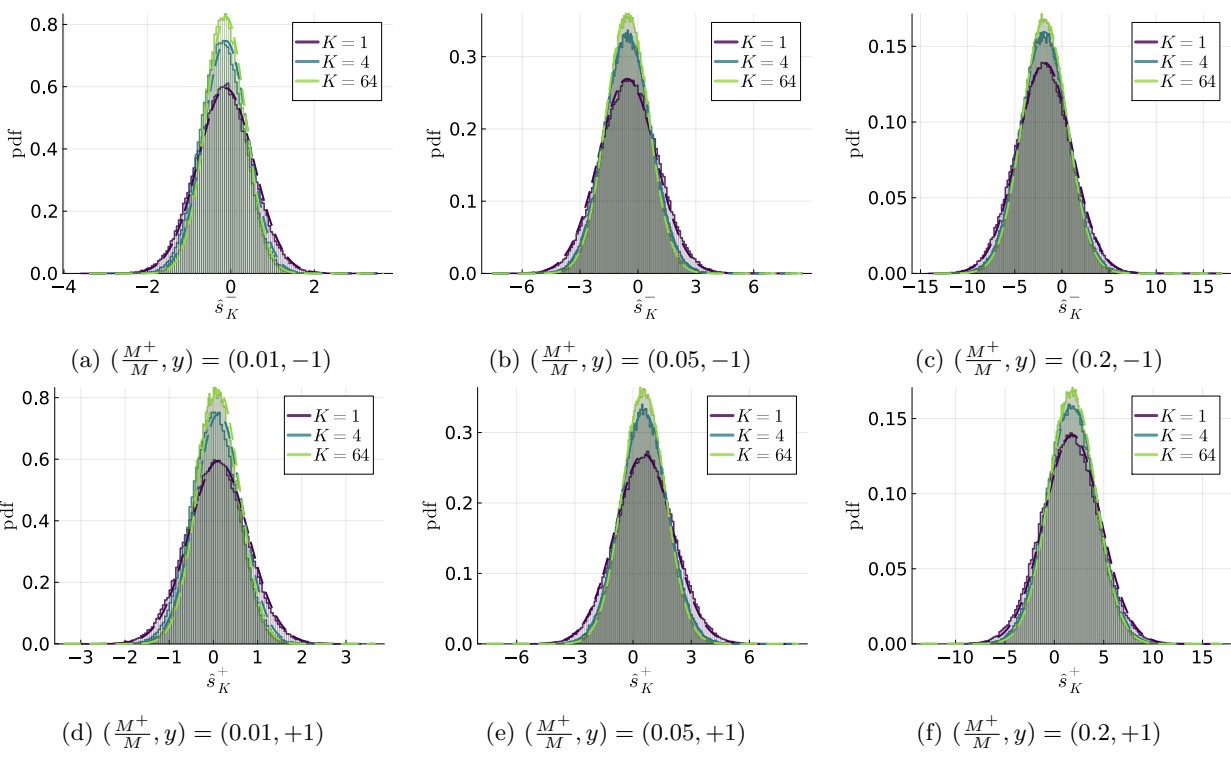

(a) $(\frac{M^+}{M}, y) = (0.01, -1)$     (b) $(\frac{M^+}{M}, y) = (0.05, -1)$     (c) $(\frac{M^+}{M}, y) = (0.2, -1)$

(d) $(\frac{M^+}{M}, y) = (0.01, +1)$     (e) $(\frac{M^+}{M}, y) = (0.05, +1)$     (f) $(\frac{M^+}{M}, y) = (0.2, +1)$

Figure 1: Comparison between the empirical distribution of $\hat{s}_K^{\pm}$ (histogram), which is obtained by a single realization of the training data $D$ of finite size with $N = 2^{13}$, and the theoretical prediction in Claim 3 (dashed). Different colors represent resampling averages with different numbers of realizations of reweighting coefficient $\boldsymbol{c}$.

$\bar{B}$ *when the bias is fixed), the prediction averaged over $K$-realization of the reweighing coefficient $\{\boldsymbol{c}_k\}_{k=1}^{K}$ when the noise term $\boldsymbol{z}$ fluctuates, is equal to that of the following random quantity:*

$$\hat{s}_K^{\pm} = B \pm m + \sqrt{q}\xi_u + \sqrt{\frac{v}{K}}\eta_u, \quad \xi_u, \eta_u \sim \mathcal{N}(0, \Delta). \tag{35}$$

The US, which uses a single realization of the resampled data set, corresponds to the case of $K = 1$.

Using the above result, one obtains the expression of the $F$-measure as follows:

**Claim 4 ($F$-measure)** *Let $(q, m, v, B)$ be the solution of the self-consistent equations in Definition 1. Then, for $\hat{s}_K^{\pm} = \frac{1}{K}\sum_{k=1}^{K} \boldsymbol{x} \cdot \hat{\boldsymbol{w}}(\boldsymbol{c}_k, D)/\sqrt{N} + \hat{B}(\boldsymbol{c}_k, D)$ (or $\bar{B}$ when the bias is fixed), the prediction averaged over $K$-realization of the reweighing coefficient $\{\boldsymbol{c}_k\}_{k=1}^{K}$, the specificity $\mathcal{S}$, the recall $\mathcal{R}$, and the $F$-measure $\mathcal{F}$ are given as follows:*

$$\mathcal{S} = H\left(-\frac{m + B}{\sqrt{\Delta(q + \frac{v}{K})}}\right), \tag{36}$$

$$\mathcal{R} = 1 - H\left(-\frac{-m + B}{\sqrt{\Delta(q + \frac{v}{K})}}\right), \tag{37}$$

$$\mathcal{F} = \frac{2}{\frac{1}{\mathcal{S}} + \frac{1}{\mathcal{R}}}, \tag{38}$$

*where $H(\cdot)$ is the complementary error function:*

$$H(x) = \mathbb{E}_{\xi \sim \mathcal{N}(0,1)}\left[\mathbb{1}(\xi > x)\right]. \tag{39}$$

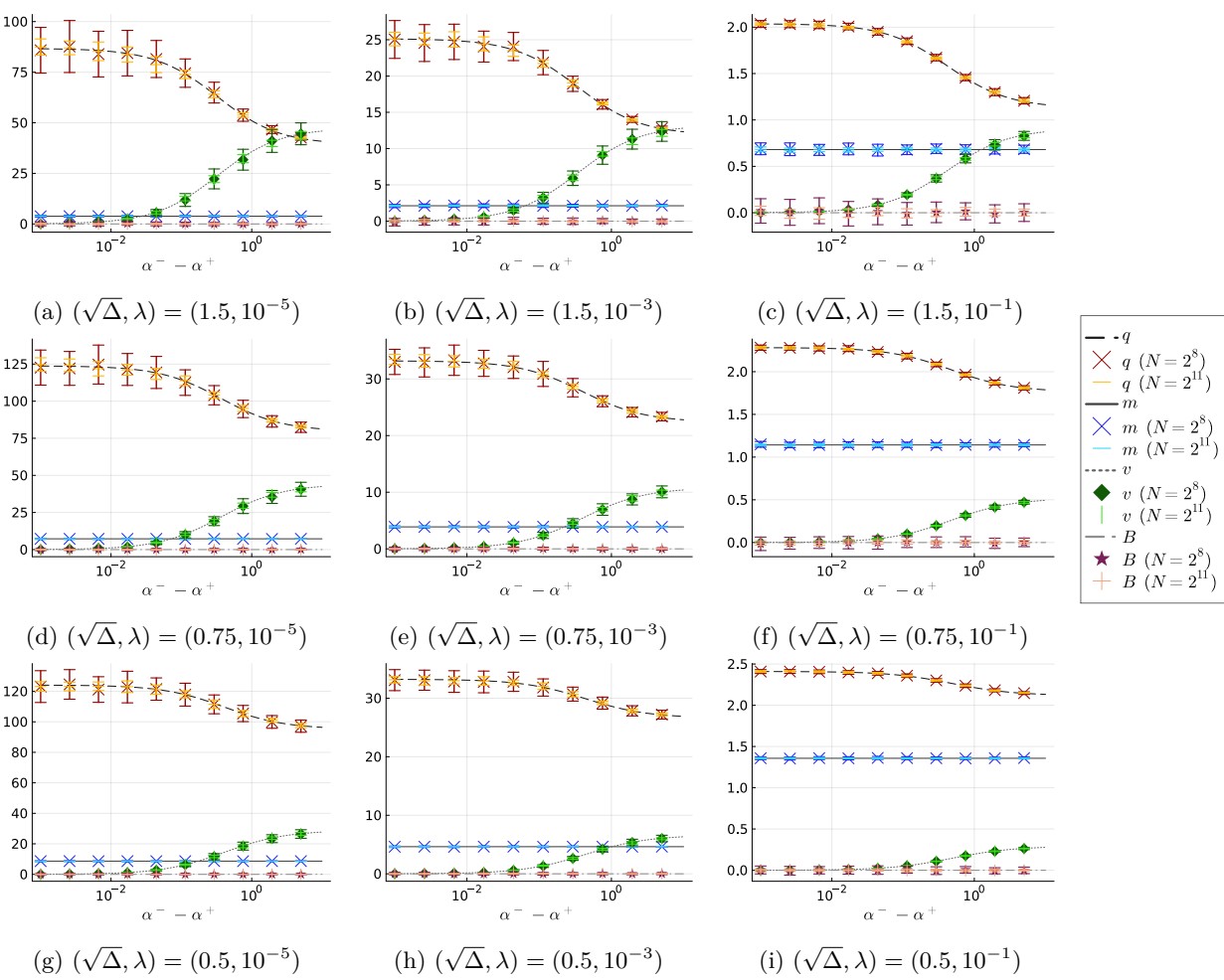

Figure 2: Comparison of the comparison of $(q, m, v, B)$, obtained as the solution of the self-consistent equation (lines), and $(N^{-1} \sum_i \mathbb{E}_{\boldsymbol{c}}[\hat{w}_i(\boldsymbol{c}, D)]^2, N^{-1} \sum_i \mathbb{E}_{\boldsymbol{c}}[\hat{w}_i(\boldsymbol{c}, D)], N^{-1} \sum_i \mathbb{V}_{\boldsymbol{c}}[\hat{w}_i(\boldsymbol{c}, D)], \hat{B}(\boldsymbol{c}, D))$ (markers with error bars). Reported experimental results are averaged over several realization of $D$ depending on the size $N$. The error bars represent standard deviations. Each panel corresponds to different values of $(\Delta, \lambda)$. Different colors of the lines represent the result for different input dimensions $N$.

We remark that for any $K \geq 1$, the variable $m$, which represents the correlation between the cluster center $\boldsymbol{v}$ and the weight vector $\hat{\boldsymbol{w}}(\boldsymbol{c}, D)$ is a constant, that is, the bagging procedure just reduces the variance of the prediction and does not improves the direction of the classification plane. An important consequence is that for any regularization parameter $\lambda > 0$, UB with $K > 1$ is a higher value of the $F$-measure when $B = 0$ and $m > 0$, indicating the superiority of UB over US.

### 3.1 Cross-checking with numerical experiments

To check the validity of Claim 3, which is the most important result for characterizing the generalization performance, we briefly compare the result of numerical solution of the self-consistent equations with numerical experiments of finite-size systems. We also compare the values of $(q, m, v, B)$ with $(N^{-1} \sum_i \mathbb{E}_{\boldsymbol{c}}[\hat{w}_i(\boldsymbol{c}, D)]^2, N^{-1} \sum_i \mathbb{E}_{\boldsymbol{c}}[\hat{w}_i(\boldsymbol{c}, D)], N^{-1} \sum_i \mathbb{V}_{\boldsymbol{c}}[\hat{w}_i(\boldsymbol{c}, D)], \hat{B}(\boldsymbol{c}, D))$ obtained by numerical experiments with finite $N$, that is, the right-hand side of equations in Claim 1 but with finite $N$.

We consider the case of subsampling (8)-(9). For the loss function, we use the cross-entropy loss; $l^+(x) = -\log(\sigma(x))$ and $l^-(x) = -\log(1 - \sigma(x))$, where $\sigma(\cdot)$ is the sigmoid function. The regularizer is the ridge:

$r(x) = x^2/2$. Also, the bias is estimated by minimizing the loss as in (7). In experiments, the scikit-learn package is used (Pedregosa et al., 2011). For simplicity, we fix the size of the overall training data as $(M^+ + M^-)/N = 1/2$. For evaluating the distribution of $\hat{s}_K^\pm$ empirically, we use $10^5$ examples of $x^\pm$.

**Predictive distributions** Figure 1 shows the comparison between the distribution in Claim 3 and the empirical distribution obtained by experiments with *single realization* of the training data $D$ of finite size $N = 2^{13}$. Different colors corresponds to different numbers of samples for the resampling average. Each panel corresponds to a different set of $(M^+/M, y, N)$, where $y$ is the label to indicate positive ($y = 1$) or negative ($y = -1$) classes. For simplicity, we fix the regularization parameter as $\lambda = 10^{-4}$, and the variance of the noise as $\Delta = 0.75^2$. In all cases, they are in good agreement, demonstrating the validity of Claim 3 and Assumption 1. See Appendix B for the result with smaller $N$.

**Parameters $\Theta$** Figure 2 shows the comparison of $(q, m, v, B)$, obtained as the solution of the self-consistent equation, and $(N^{-1}\sum_i \mathbb{E}_c[\hat{w}_i(c, D)]^2, N^{-1}\sum_i \mathbb{E}_c[\hat{w}_i(c, D)], N^{-1}\sum_i \mathbb{V}_c[\hat{w}_i(c, D)], \hat{B}(c, D))$. The values are plotted as functions of $\alpha^- - \alpha^+$, the difference of the normalized size of the majority and the minority classes. Each panel corresponds to a different set of $(\lambda, \sqrt{\Delta})$. For numerical experiments, two different input dimensions $N = 256(= 2^8)$, and $2048(= 2^{11})$ are used. To take the average over $c$, 128 independent realizations are used. Reported experimental results are averaged over several realization of $D$ depending on the size $N$. The results show that the theoretical predictions and the experimental values are in good agreement. Also, the error bars represent standard deviations that decrease as $N$ grows, indicating the concentration of these quantities. Overall, the figure confirms the validity of Claim 1.

## 4 Under-bagging and regularization

In this section, using Claim 4, we compare the performance of several estimators, that is, US, UB, and SW. In the following, we consider the limit $K \to \infty$ in (36)-(37), which yields the smallest variance.

For simplicity, we focus on the case of ridge-regularized cross-entropy loss as in subsection 3.1. As a resampling method, we mainly focus on the sampling without replacement defined as in (8)-(9), except for subsection 4.3. When sampling without replacement is used, we can expect that the estimated bias term $\hat{B}(c, D)$ to be zero, since the number of positive and negative samples is balanced. See also Figure 2. Therefore, we set the bias to zero a priori when using sampling without replacement. Otherwise, the bias is estimated as in (7). Furthermore, as investigated in the previous studies (Dobriban & Wager, 2018; Mignacco et al., 2020), when training a linear classifier using balanced data and a ridge-regularized cross-entropy loss, an infinitely large regularization parameter yields the best direction for the classification plane. Hence we obtain infinitely large regularization parameter if we optimize the regularization parameter. However, as reported in (Mignacco et al., 2020; Takahashi, 2024), the discrepancy between the theory and experiment with finite $N$ can be huge if too a large regularization parameter is used because the norm of the weight vector can be pathologically small. Therefore we consider several finite and fixed regularization parameters instead of optimizing them[3].

### 4.1 US and UB

In this subsection, we compare the performance of US and UB, through the values of $F$-measures.

Figure 3 shows the dependence of the $F$-measure $\mathcal{F}$ on $\alpha^- - \alpha^+$, which is the difference in the scaled size between the majority (negative) and minority (positive) classes, or equivalently, the scaled size of the excess majority examples. Each panel corresponds to different values of the variance $\Delta$ and the regularization parameter $\lambda$. Recall that the $F$-measure is uppder bounded by unity. In UB, the $F$-measure increases monotonically as the number of excess data points in the majority class $\alpha^- - \alpha^+$ increases, despite of the large class imbalance. However, in the case of US, the $F$-measure does not depends on $\alpha^- - \alpha^+$, i.e., it is not affected by the class imbalance, but cannot utilize the excess data points in the majority class. This

---

[3]If we consider non-symmetric Gaussian cluster setups, the optimal regularization parameter would be non-trivial values, but this is beyond the scope of this work, and we leave it here as a future work.

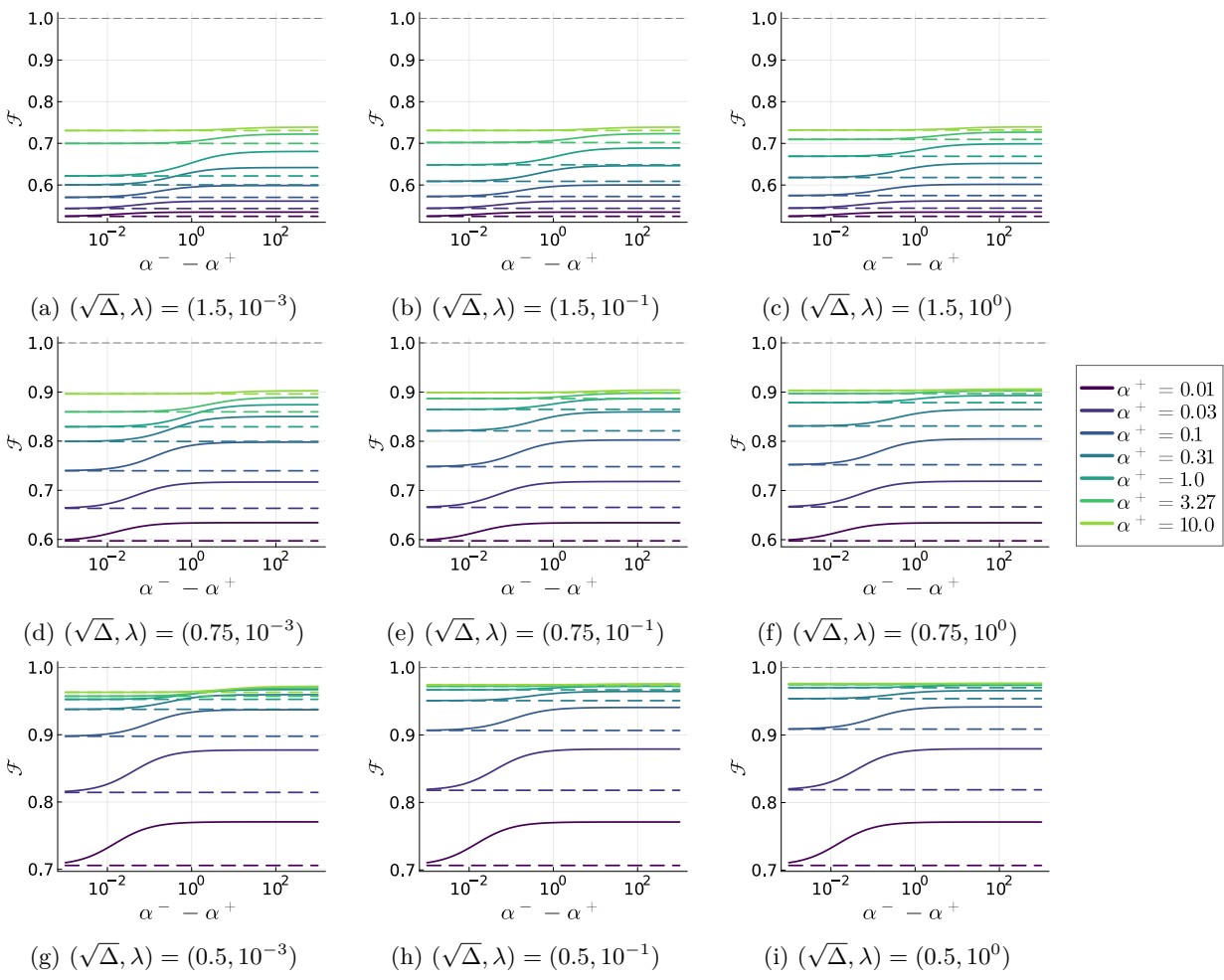

Figure 3: Comparison of the dependence of the $F$-measure $\mathcal{F}$ on $\alpha^- - \alpha^+$, which is the difference in the size between the majority (negative) and the minority (positive) classes, between US (dashed) and UB (solid). Each panel corresponds to different values of $(\Delta, \lambda)$. Different colors of the lines represent the result for different sizes of the minority classes $\alpha^+$.

tendency does not depend on the values of $(\Delta, \lambda)$. Actually, when subsampling is used as the resampling method, one can show that the performance of the classifier does not depends on $\alpha^- - \alpha^+$ as shown in Appendix C. Intuitively, this is because considering uniform subsampling without replacement essentially is just using a smaller size of negative examples.

Figure 4 shows heatmap of the relative $F$-measure $\mathcal{F}_{\mathrm{UB}}/\mathcal{F}_{\mathrm{US}}$, where $\mathcal{F}_{\mathrm{UB}}$ and $\mathcal{F}_{\mathrm{US}}$ are the $F$-measure for UB and US, respectively. It is clear that the values of the $F$-measures are improved when the variance is small, the size of the minority class is small and the excess size of the majority class is large. When the variance is large, the improvement at extremely small $\alpha^+$ region is small because $m$ in (38) can be small due to the large overlap between two clusters. We remark that the boundary where data is linearly separable/inseparable is roughly in the region of $\alpha^+ \in (10^0, 10^1)$ (see Figure 5). Therefore, the large improvement in generalization performance below this boundary indicates that UB is effective in the overparameterized settings, as indicated in the literature (Wallace et al., 2011).

### 4.1.1 Double-descent-like behavior

It is also noteworthy that the improvement may be large around some values of $\alpha^+$ when the regularization parameter is extremely small. This is because the variance associated with random reweighting coefficient

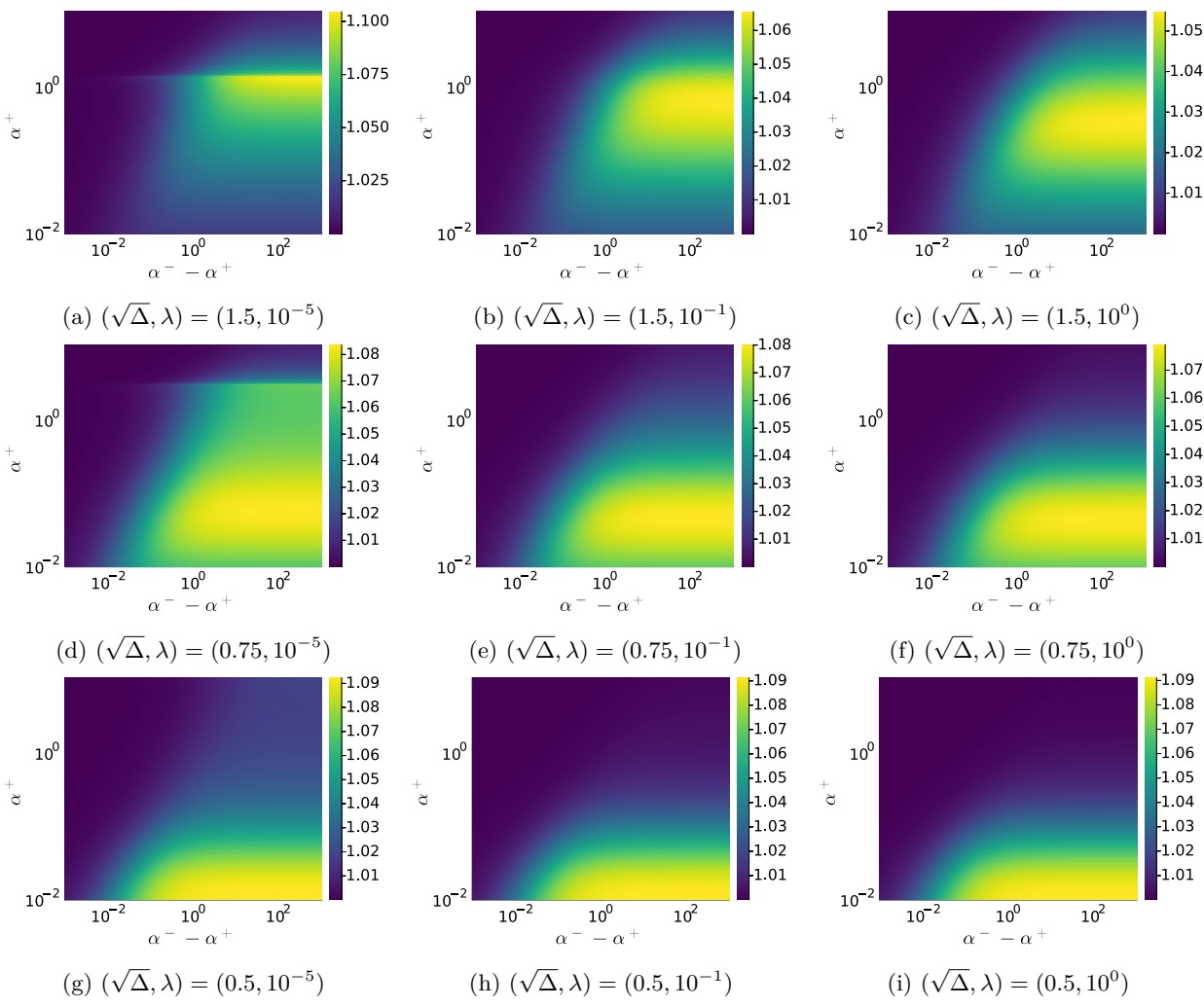

Figure 4: Heatmap plot for the relative $F$-measure $\mathcal{F}_{\mathrm{UB}}/\mathcal{F}_{\mathrm{US}}$, where $\mathcal{F}_{\mathrm{UB}}$ and $\mathcal{F}_{\mathrm{US}}$ are the $F$-measure for UB and US, respectively. Each panel corresponds to different values of $(\Delta, \lambda)$.

*c* exhibits double-descent-like behavior with respect to $\alpha^+$ and takes a large value at a certain $\alpha^+$. This point is evident in Figure 5 that shows the relative variance $v/(q + v)$ for several values of $\Delta$. It is clear that the relative variance takes a peak value at the interpolation threshold obtained by the formula in the logistic regression with balanced two-component Gaussian mixture with data size $\alpha^+ N$ (Mignacco et al., 2020). Similar double-descent-like behavior is also observed in an analysis of bagging without class imbalance structure (Clarté et al., 2024). This indicates that the UB, which can remove the contribution from $v$, is robust to the existence of the phase transition from linearly separable to inseparable training data, although the performance of standard interpolator obtained by a single realization of training data is affected by such a phase transition as reported in (Mignacco et al., 2020).

## 4.2 UB and SW

In this section, we compare the performance of UB with SW, through the values of $F$-measure. For SW, we consider two types of weighting coefficients $\gamma^\pm$. The first choice is $(\gamma^+, \gamma^-) = (\gamma^+_{\mathrm{naive}}, \gamma^-_{\mathrm{naive}}) \equiv ((M^+ + M^-)/(2M^+), (M^+ + M^-)/(2M^-))$, which is a naive guess often used as the default value of the major ML packages such as the scikit-learn (King & Zeng, 2001; Pedregosa et al., 2011). Another scenario considers the case where the weighting coefficients are optimized so as to maximize the $F$-value using the Nelder-

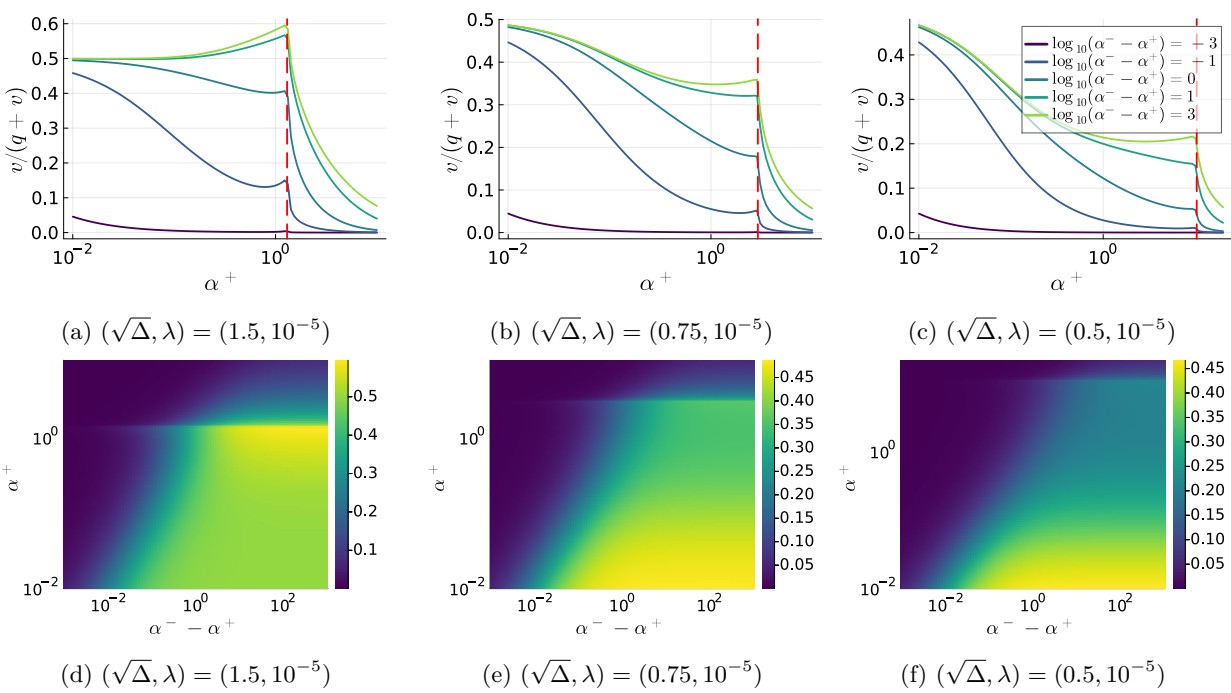

Figure 5: Relative variance $v/(q + v)$. Upper panels: the relative variance shown as a function of $\alpha^+$ for some selected values of $\alpha^- - \alpha^+$. The legend is common across all panels. Red dashed lines represent the interpolation threshold. Lower panels: the relative variance shown as a 2d-heatmap.

Mead method in Optim.jl package (Mogensen & Riseth, 2018). In order to make a fair comparison with the UB with the regularization strength $\lambda$, the regularization parameter for the SW with optimal weighting coefficient is optimized in the range below $\lambda_{\mathrm{US}} \leq (\gamma_{\mathrm{naive}}^+ + \gamma_{\mathrm{naive}}^-)\lambda$. For simple weighting with the naive coefficients, the regularization parameter is just fixed as a constant value.

Figure 6 shows the comparison of the $\alpha^- - \alpha^+$ dependence of the $F$-measure between UB and SW for selected values of $\alpha^+$. It is clear that the performance of UB (solid line) and SW with the optimized coefficients (dashed line) are comparable in all cases. This is more clearly evident in the heatmap in Figure 7, which shows that the relative $F$-measure $(\mathcal{F}_{\mathrm{UB}} - \mathcal{F}_{\mathrm{SW}})/\mathcal{F}_{\mathrm{UB}}$ is within about $1 \pm 10^{-2}$. On the other hand, the performance of the SW method with the naive weighting coefficients $\gamma_{\mathrm{naive}}^{\pm}$ (dash-dot) is catastrophically worse than UB where $\alpha^+$ is small and $\alpha^- - \alpha^+$ is large. As $\alpha^- - \alpha^+$ increases, the $F$-measure decreases monotonically. See also Figure 13 in Appendix D for a heatmap view. Figure 8 shows the ratio between the optimal and naive weighting coefficients. It is clear that when the variance $\Delta$ of the cluster is large, the coefficient for the majority class (dash) should be reduced by an order of magnitude from the naive value, indicating that a delicate tuning of the weighting coefficients is needed for the SW method.

These results indicate that the combination of the weighting and regularization can be similar to the combination of the under-sampling and bagging, although UB does not require delicate adjustment of the resampling rate.

### 4.2.1 UB and SW in real-world dataset

Unfortunately, it is unclear whether the above similarity between bagging with class-dependent resampling and weighting generally holds beyond Gaussian data. However, to verify the universality of the similarity between SW with the optimal coefficients and UB, we show here the results of a simple experiment using Fashion-MNIST data (Xiao et al., 2017), which consists of $N = 28^2 = 784$ dimensional images from 10 classes. For this, we performed binary classification with the logistic regression using the classes "T-shirt/top" and "Shirt". Specifically, we chose the "T-shirt/top" class as a positive class and extracted $M^+ = 50$ data

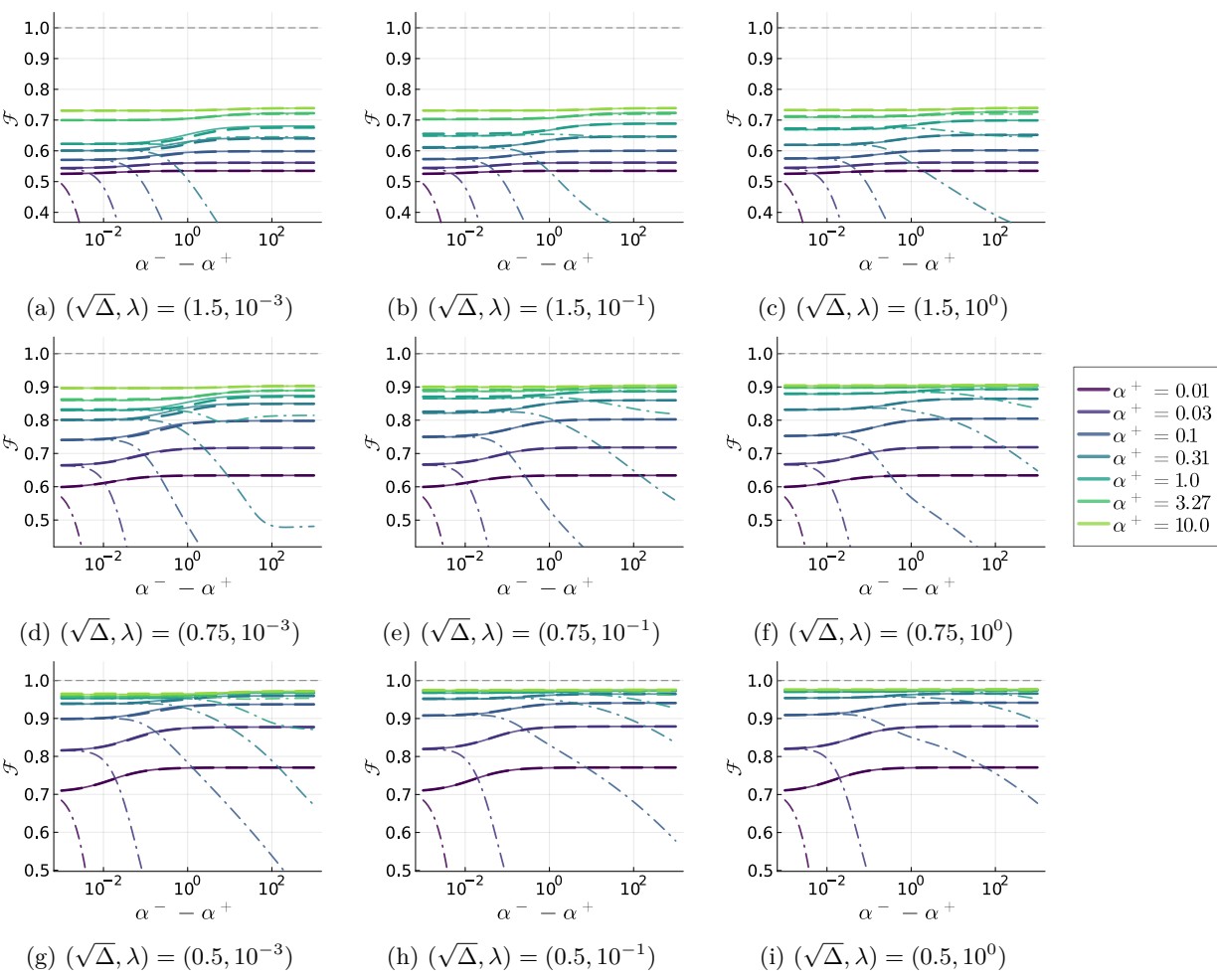

Figure 6: The comparison of the dependence of the $F$-measure $\mathcal{F}$ on $\alpha^- - \alpha^+$, which is the difference in the size between the majority (negative) and the minority (positive) classes, between weighting with optimal coefficients(dashed), weighting with naive coefficients (dashdot), and UB (solid). Each panel corresponds to different values of $(\Delta, \lambda)$, where $\lambda$ is the regularization parameter. Different colors of the lines represent the result for different sizes of the minority classes $\alpha^+$.

points. Then, the size of the "Shirt" class, which was specified as negative, was varied from $M^- = 51$ to 3000 to check the behavior of the $F$-measure and the weighting coefficients. The weighting coefficients were optimized using validation data with 400 images, and the $F$-measure was computed using test data with 1600 images.

Figure 9 shows the results of the experiment. The behavior of the $F$-measure and the weighting coefficients are similar to those observed for Gaussian mixture data in Section 4.1. The performance of SW with the naive coefficients drops rapidly with the number of excess majority samples. However, the performance of SW with the optimal coefficients is similar to that of UB. Moreover, the optimal weighting coefficients for the majority class are several orders of magnitude smaller than the naive coefficients, although the statistical fluctuation is large.

Of course, a more thorough experiment or theoretical proof is needed to fully verify the universality of the above results, but this is beyond the scope of this study and is left as a future work.

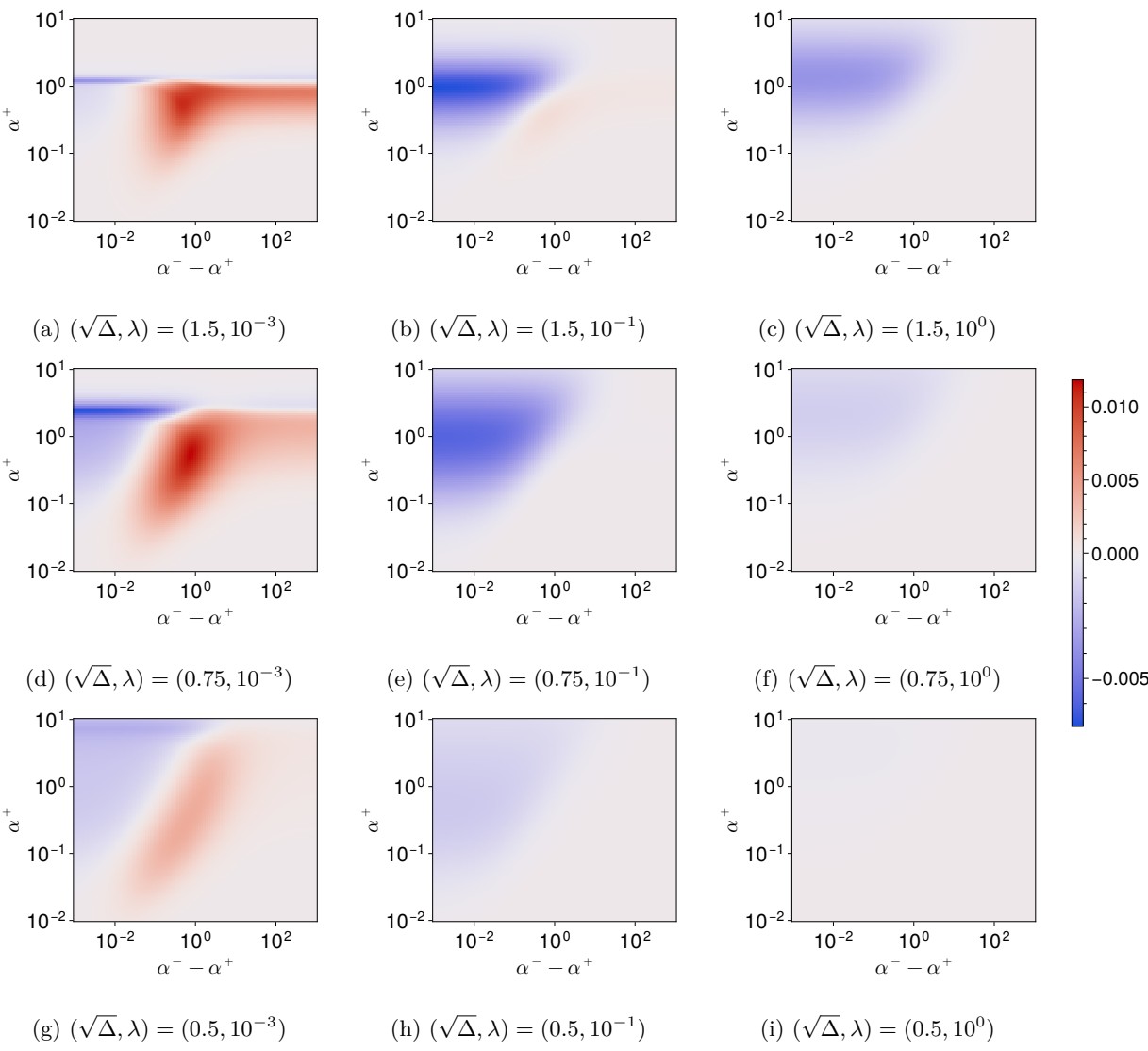

Figure 7: The heatmap plot for the relative $F$-measure: $(\mathcal{F}_{\mathrm{UB}} - \mathcal{F}_{\mathrm{SW}})/\mathcal{F}_{\mathrm{UB}}$, where $\mathcal{F}_{\mathrm{UB}}$ and $\mathcal{F}_{\mathrm{SW}}$ are the $F$-measure for UB and the weighting method, respectively. Each panel corresponds to different values of $(\Delta, \lambda)$, where $\lambda$ is the regularization parameter used in UB.

## 4.3 Sampling with and without replacement

In this section, we compare the behavior of the estimators obtained by sampling with (10)-(11) replacement and without (8)-(9) replacement. The difference between these resampling methods is mainly whether or not the resampled data contains multiple instances of the same data point. We are interested in whether this makes a significant performance difference in terms of the $F$-measure.

Unlike sampling without replacement, which we have considered so far, the expression for the optimal resampling rate cannot be obtained explicitly even in a naive sense. In this paper, motivated by the fact that the $F$-measure considered here treats the classification error equally for positive and negative samples, we numerically find the resampling rate such that the bias term is estimated to be zero. As shown in Appendix E, the $F$-measure actually takes the highest value at this resampling rate. Moreover, we focus on the case of $\Delta = 0.75^2$, since it is expected from the results of sampling without replacement so far that the qualitative behavior of the $F$-measure does not strongly depend on the value of $\Delta$.

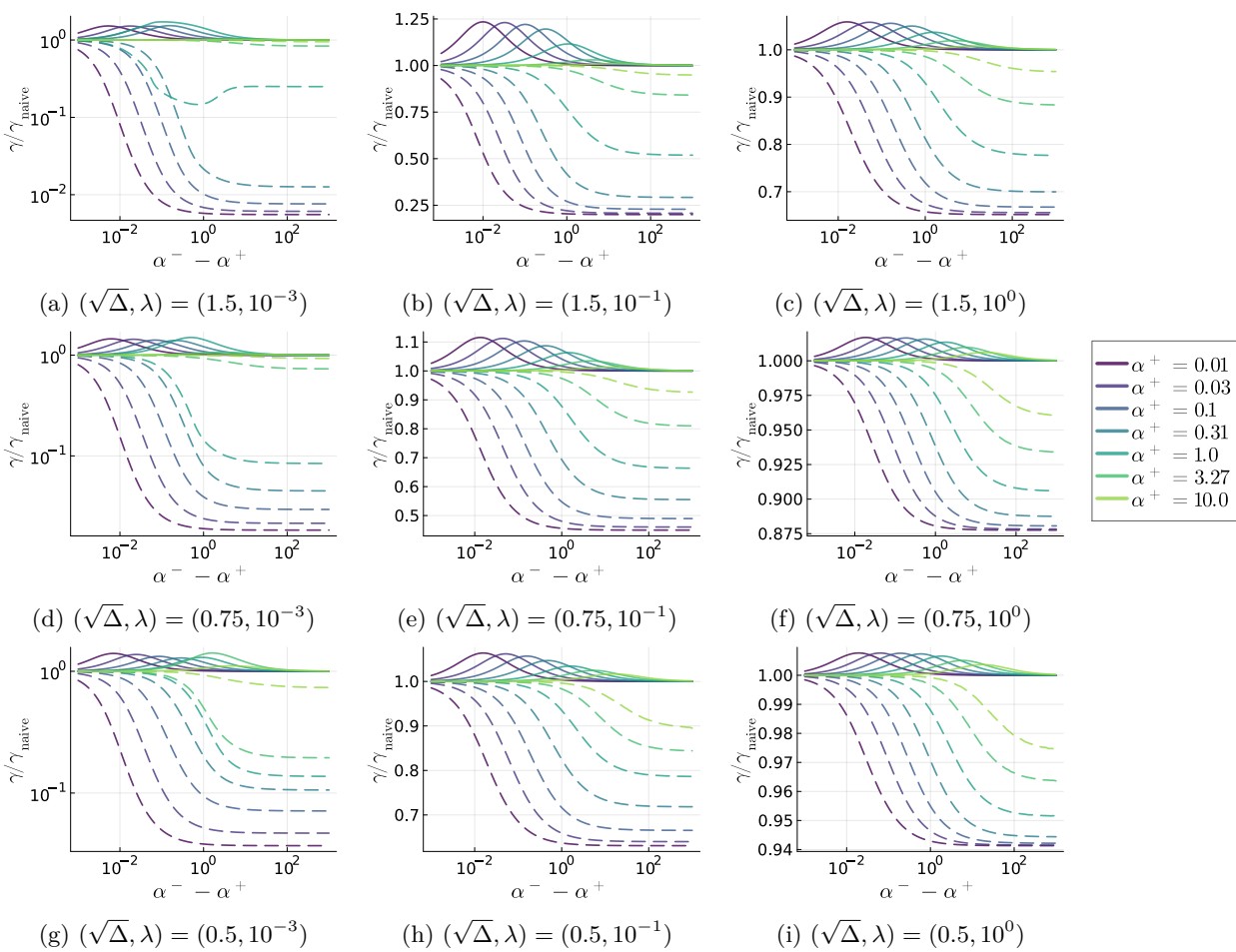

Figure 8: The comparison of the weighting coefficients $\gamma^{\pm}/\gamma^{\pm}_{\mathrm{naive}}$ as a function of $\alpha^{-} - \alpha^{+}$. The ratio for the majority class is shown by dashed lines, and that for the minority class is shown by solid lines. Each panel corresponds to different values of $(\Delta, \lambda)$, where $\lambda$ is the regularization parameter for UB. Different colors of the lines represent the result for different sizes of the minority classes $\alpha^{+}$.

Figure 10 shows the comparison of the $F$-measures obtained by sampling with and without replacement. Panels (a)-(c) show the $F$-measures for the UB estimators obtained by each resampling method. It is clear that the predictive performance of the UB estimator shows little difference between sampling with and without replacement in terms of the $F$-measure. In particular, they agree almost perfectly for $\alpha^{-} - \alpha^{+} \gg 1$, because the resampling rate $\mu_{*}$ for sampling with replacement is small in this region and there are almost no duplicate data points. On the other hand, for $\alpha^{-} - \alpha^{+} \ll 1$, sampling with replacement shows slightly better performance than sampling without replacement, because sampling with replacement induces more variation in each of the resampled data, and the regularization effect of ensembling can be stronger. Recall that each resampled data set produced by sampling without replacement is nearly identical when $\alpha^{-} - \alpha^{+} \ll 1$. Also, as shown in panels (d)-(f), the performance of the US estimator depends slightly on $\alpha^{-} - \alpha^{+}$. This is because the size of the resampled majority class data is not constant due to the effect of duplication of the data points.

Figure 11 shows the resampling rate, which is the expected size of the resampled majority class data, to make the bias term zero. As expected, there is almost no difference between the two resampling methods for $\alpha^{-} - \alpha^{+} \gg 1$ since the effect of duplication is quite small. However, the resampling rate of sampling with replacement is larger than the other for $\alpha^{-} - \alpha^{+} \ll 1$, because a large resampling rate is required to

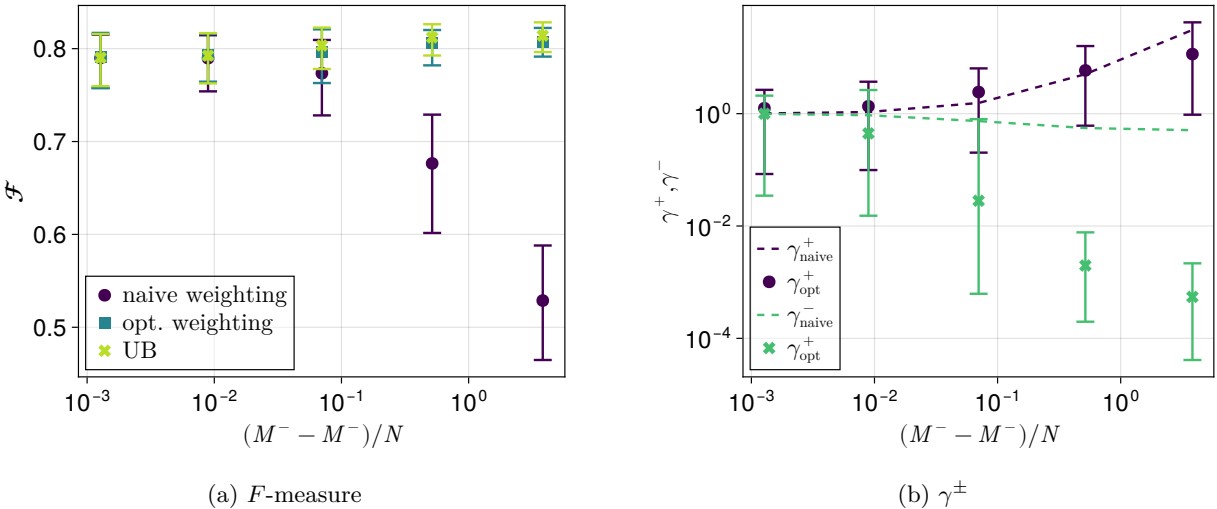

(a) $F$-measure

(b) $\gamma^{\pm}$

Figure 9: F-measure and weighting coefficients in the Fashion-MNIST data. Markers with error bars represent the experimental results. Dashed lines represent the naive reweighting coefficients: $\gamma^{\pm} = (M^+ + M^-)/(2M^{\pm})$. Here, $N = 28^2 = 784$. The regularization parameter is set to $\lambda = 10^{-3}$. Error bars are constructed from 2.5 and 97.5 percentile points over 64 independent random train and validation splits.

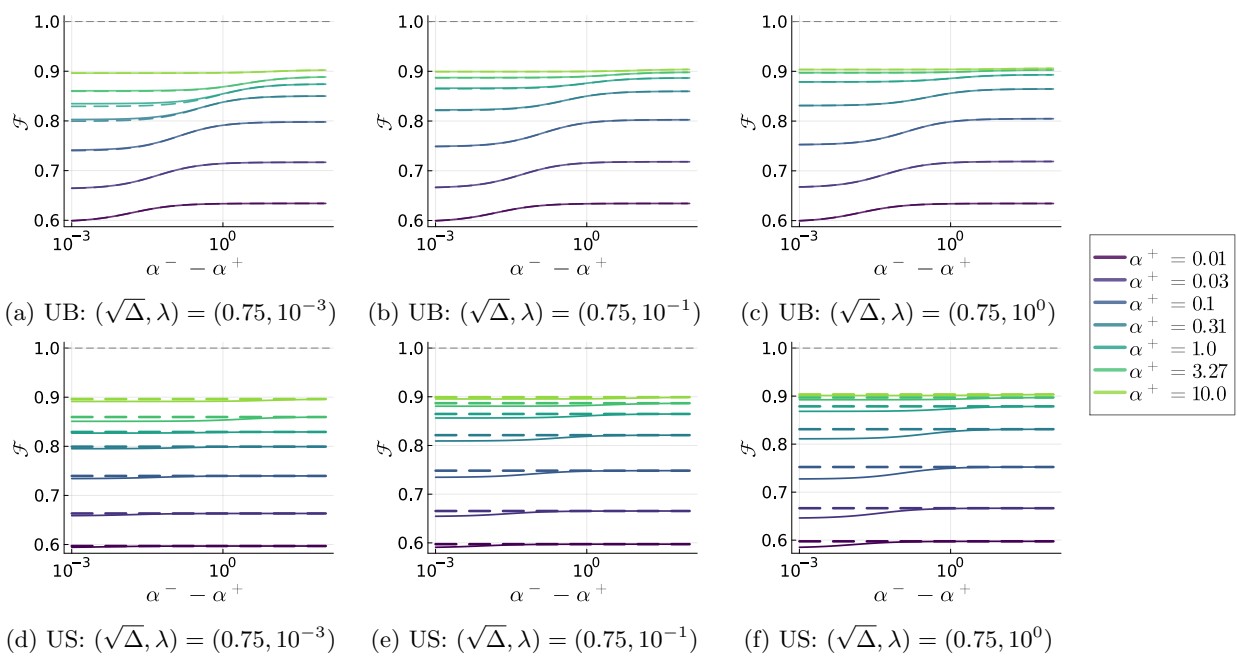

(a) UB: $(\sqrt{\Delta}, \lambda) = (0.75, 10^{-3})$  (b) UB: $(\sqrt{\Delta}, \lambda) = (0.75, 10^{-1})$  (c) UB: $(\sqrt{\Delta}, \lambda) = (0.75, 10^{0})$

(d) US: $(\sqrt{\Delta}, \lambda) = (0.75, 10^{-3})$  (e) US: $(\sqrt{\Delta}, \lambda) = (0.75, 10^{-1})$  (f) US: $(\sqrt{\Delta}, \lambda) = (0.75, 10^{0})$

Figure 10: The comparison of the dependence of the $F$-measure $\mathcal{F}$ on $\alpha^- - \alpha^+$, which is the difference in the size between the majority (negative) and the minority (positive) classes, between sampling with (solid) and without (dashed) replacement. Panels (a)-(c): the comparison between UB estimators. Panels (d)-(e): the comparison between the US estimators. The variance is fixed to $\sqrt{\Delta} = 0.75$.

include all majority class data points in each resampled data, suggesting that sampling without replacement is superior in terms of computational efficiency.

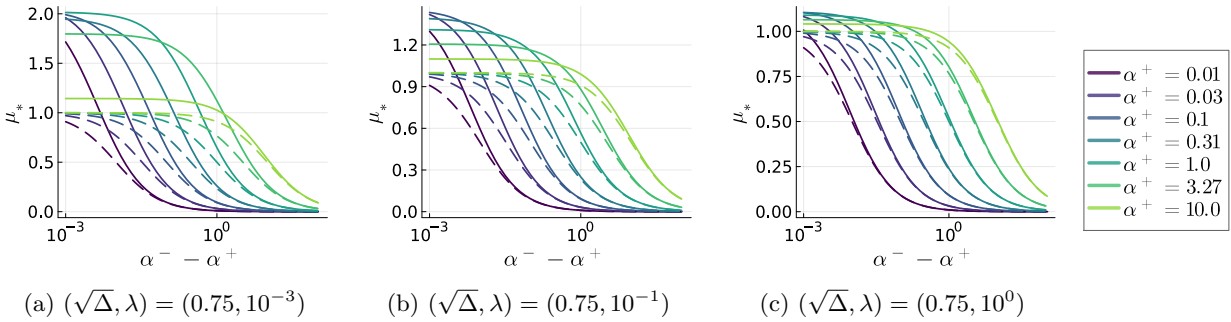

Figure 11: The comparison of the dependence of the optimal resampling rate $\mu_*$ on $\alpha^- - \alpha^+$ between sampling with (solid) and without (dashed) replacement. The variance is fixed to $\sqrt{\Delta} = 0.75$.

## 5 Summary and Discussions

In this work, we derived a sharp asymptotics of the estimators obtained by the randomly reweighted losses (7) in the limit where the data size and the input dimension diverge proportionally. The derivation was based on the standard use of the replica method. The results are summarized in Claim 1-5.

Using the derived sharp asymptotics, we investigated the performance of UB, US, and SW. Our main findings were (i) UB can improve the performance of the classifier in terms of the $F$-measure by increasing the size of the majority class while keeping the size of the minority class fixed, even when the degree of class imbalance becomes large, (ii) the performance of the classifier obtained by US was almost independent of the size of the excess majority examples, and (iii) the performance of the SW method was almost equivalent to UB if the weighting coefficients are carefully optimized, but if not properly adjusted, the performance drops rapidly as the number of the excess majority examples increases. (iv) Furthermore, UB was robust to the interpolation phase transition in contrast to US. In conclusion, ensembling and ridge regularization are different when using under-sampled data. This result is different from the case of naive bagging in training GLMs without considering the class imbalance structure, where ensembling with uniform resampling rate and the simple ridge regularization give exactly the same performance. Rather, UB is similar to the combination of the weighting method with optimal weighting coefficients and the ridge regularization. These result may indicate that one can obtain performance improvements considering class-dependent resampling or reweighting in learning classifiers from imbalanced data in contrast to the past results (Nixon et al., 2020).

Although UB is a powerful method for learning in imbalanced data, it requires an increased computational cost proportional to the number of under-sampled datasets, which can be a serious problem in learning deep neural networks. Therefore, the development of simple heuristics to achieve similar results as UB would be a promising future direction.

### Acknowledgments

This study was supported by JSPS KAKENHI Grant No. 21K21310 and 23K16960, and Grant-in-Aid for Transformative Research Areas (A), "Foundation of Machine Learning Physics" (22H05117).

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

| Notation | Description |
|---|---|
| $\text{extr}_x f(x)$ | extremization with respect to $x$ |
| $d\boldsymbol{x}$ | with $\boldsymbol{x} \in \mathbb{R}^N$, a measure over $\mathbb{R}^N$ |
| $d^n \boldsymbol{x}$ | with $\boldsymbol{x}_1, \dots, \boldsymbol{x}_n \in \mathbb{R}^N$, a measure over $\mathbb{R}^{N \times n}$ |
| $I_N$ | identity matrix of size $N \times N$ |
| $\partial^k \mathcal{F}(y)$ | for a function $\mathcal{F}(x)$ and integer $k$, |
| | the partial derivative of $\mathcal{F}$ with respect to its argument at $x = y$: $\left. \frac{\partial^k \mathcal{F}(x)}{\partial x^k} \right|_{x=y}$ |
| $f(n) = \mathcal{O}(g(n))$ | Landau's O as $n \to 0$; $\lvert f(n)/g(n) \rvert < \infty$ as $n \to 0$ |

Table 2: Additional notations used in the appendix

# A    Derivation

In this appendix, we outline the derivations of Claim 2. The derivation is based on the standard use of the replica method of statistical physics (Mézard et al., 1987; Charbonneau et al., 2023; Montanari & Sen, 2024). For a pedagogical introduction to the replica method computation for the linear models, see literature (Krzakala & Zdeborová, 2021), for example.

As noted in the related works section, the salient feature of the asymptotics in the LSL is that the macroscopic properties of the learning results, such as the predictive distribution, typically do not depend on each realization of the training data. In this paper, we assume these concentration properties in advance to derive the formulas for the asymptotic properties.

**Assumption 1 (Concentration of the macroscopic properties of the learning results)** *We     assume that the macroscopic properties of the estimators (7) will concentrate at LSL. Concretely, we assume that the values of $(N^{-1} \sum_i \mathbb{E}_{\boldsymbol{c}}[\hat{w}_i(\boldsymbol{c}, D)]^2, N^{-1} \sum_i \mathbb{E}_{\boldsymbol{c}}[\hat{w}_i(\boldsymbol{c}, D)], N^{-1} \sum_i \mathbb{V}_{\boldsymbol{c}}[\hat{w}_i(\boldsymbol{c}, D)], \hat{B}(\boldsymbol{c}, D))$ does not fluctuate with respect to the realization of $D$ at LSL. Also, Let $\boldsymbol{x}^{\pm} = \pm \boldsymbol{v}/\sqrt{N} + \boldsymbol{z}$, $z_i \sim_{\text{iid}} p_z, i \in [N]$ be a new input. We assume that the distribution of the prediction for this new input $\hat{s}^{\pm}(\boldsymbol{c}, D) = \boldsymbol{x}^{\pm} \cdot \hat{\boldsymbol{w}}(\boldsymbol{c}, D)/\sqrt{N} + \hat{B}(\boldsymbol{c}, D)$ (or $\bar{B}$ when the bias is fixed) with respect to $\boldsymbol{c}$ and $\boldsymbol{z}$ does not fluctuate with respect to the realization of $D$ at LSL.*

## A.1    Additional notations

To describe the derivations, we use some additional shorthand notations. We summarize them in Table 2.

## A.2    Boltzmann distribution

Let us starting with rewriting the training (7) as a statistical physics problem. For this, we introduce a probability density function $p_{\text{B}}$, which is termed the *Boltzmann distribution*, as follows. Let $\mathcal{L}(\boldsymbol{\theta}; D, \boldsymbol{c})$ be the cost function to be optimized in (7). Then, the Boltzmann distribution is defined as

$$p_{\text{B}}(\boldsymbol{\theta}|D, \boldsymbol{c}) = \frac{1}{Z(D, \boldsymbol{c})} e^{-\beta \mathcal{L}(\boldsymbol{\theta}; D, \boldsymbol{c})}, \tag{40}$$

where $\beta \in (0, \infty)$ is the positive parameter termed the inverse temperature following the custom of statistical physics, and $Z(D, \boldsymbol{c}) = \int e^{-\beta \mathcal{L}(\boldsymbol{\theta}; D, \boldsymbol{c})} d\boldsymbol{\theta}$ is the normalization constant. For simplicity, we use the same notation for the model's parameter whether the bias is fixed or not. We basically consider the case when the bias is estimated, but just replace $B$ with $\bar{B}$ when the bias is fixed.

At $\beta \to \infty$, the Boltzmann distribution converges to the uniform distribution over the minimum of $\mathcal{L}(\boldsymbol{\theta}; D, \boldsymbol{c})$. Since we are considering a convex loss function, the Boltzmann distribution converges to the Delta function at $\beta \to \infty$:

$$\lim_{\beta \to \infty} p_{\text{B}}(\boldsymbol{\theta}|D, \boldsymbol{c}) = \delta_{\text{d}}(\boldsymbol{\theta} - \hat{\boldsymbol{\theta}}(D, \boldsymbol{c})). \tag{41}$$

Therefore, analyzing the Boltzmann distribution at this limit is equivalent to analyzing the estimator defined in (7).

### A.3 Distribution of the prediction

In the following, we use the replica method to consider the distribution of the prediction for a new input.

#### A.3.1 Replicated system

We are interested in how the prediction for a new input $\hat{s}^{\pm}(\boldsymbol{c}, D) \equiv \boldsymbol{x}^{\pm} \cdot \hat{\boldsymbol{w}}(\boldsymbol{c}, D)/\sqrt{N} + \hat{B}(\boldsymbol{c}, D)$ depends on $\boldsymbol{x}$ and $\boldsymbol{c}$. For this, we introduce another probability density, which we refer to as the *replciated system*, as follows. Let $n_1, n_2 \in \mathbb{N}$ be the positive integers. Then, the replicated system is defined as follows:

$$\tilde{p}(\{\boldsymbol{\theta}_{\boldsymbol{a}}\}) = \frac{1}{\Xi} \mathbb{E}_D \left[ \prod_{a_1=1}^{n_1} \mathbb{E}_{\boldsymbol{c}_{a_1} \sim p_c} \left[ \prod_{a_2=1}^{n_2} e^{-\beta \mathcal{L}(\boldsymbol{\theta}_{\boldsymbol{a}}; D, \boldsymbol{c}_{a_1})} \right] \right], \tag{42}$$

where $\boldsymbol{a} = (a_1 a_2)$ is the shorthand notation for the subscript, and $\Xi$ is the normalization constant. This is the density over $\mathbb{R}^{(N+1)\times(n_1\times n_2)}$ if the bias is trained and $\mathbb{R}^{N\times(n_1\times n_2)}$ if the bias is fixed. Recall that the replicated system is not conditioned by $D$ or $\boldsymbol{c}$ since the average is taken explicitly.

Let $l = 1, 2, \ldots$ be the positive integer, and take $n_1 > l$. Also, let $\boldsymbol{a}^{(1)} = (a_1^{(1)} a_2^{(1)}), \ldots, \boldsymbol{a}^{(l)} = (a_1^{(l)} a_2^{(l)})$ be a set of indices such that $a_1^{(k)} \neq a_1^{(k')}$ for any $k \neq k' \in [l]$. Then, for an arbitrary function $\psi(\cdot)$ such that subsequent integrals converge, we consider the following expectation:

$$\Omega_{l,n_1,n_2}^{\pm} \equiv \mathbb{E}_{\{\boldsymbol{\theta}_{\boldsymbol{a}}\} \sim \tilde{p}, \boldsymbol{x}^{\pm}} \left[ \prod_{k=1}^{l} \psi(\boldsymbol{x}^{\pm} \cdot \boldsymbol{w}_{\boldsymbol{a}^{(k)}}/\sqrt{N} + B_{\boldsymbol{a}^{(k)}}) \right]. \tag{43}$$

By formally taking the integral with respect to $\{\boldsymbol{\theta}_{\boldsymbol{a}}\}$, we obtain the following:

$$\Omega_{l,n_1,n_2}^{\pm} = \frac{1}{\mathbb{E}_D \left[ \mathbb{E}_{\boldsymbol{c}}[Z(D,\boldsymbol{c})^{n_2}]^{n_1} \right]} \mathbb{E}_{D,\boldsymbol{x}^{\pm}} \left[ \mathbb{E}_{\boldsymbol{c}} \left[ \mathbb{E}_{\boldsymbol{\theta} \sim p_{\mathrm{B}}} \left[ \psi(\boldsymbol{x}^{\pm} \cdot \boldsymbol{w}/\sqrt{N} + B) \right] Z(D,\boldsymbol{c})^{n_2} \right]^l \mathbb{E}_{\boldsymbol{c}} \left[ Z(D,\boldsymbol{c})^{n_2} \right]^{n_1-l} \right]. \tag{44}$$

Since the above expression depends on $n_1$ and $n_2$ in the form of the power function, we can consider formal analytical continuation of $n_1, n_2 \in \mathbb{N}$ to $n_1, n_2 \in \mathbb{R}$, and take the limit $n_1, n_2 \to 0$. Then, we obtain the following expression:

$$\lim_{n_1,n_2 \to 0} \Omega_{l,n_1,n_2}^{\pm} = \mathbb{E}_{D,\boldsymbol{x}^{\pm}} \left[ \mathbb{E}_{\boldsymbol{c}} \left[ \mathbb{E}_{\boldsymbol{\theta} \sim p_{\mathrm{B}}} \left[ \psi(\boldsymbol{x}^{\pm} \cdot \boldsymbol{w}/\sqrt{N} + B) \right] \right]^l \right]. \tag{45}$$

At the limit $\beta \to \infty$, the above converges to the following:

$$\lim_{\beta \to \infty} \lim_{n_1,n_2 \to 0} \Omega_{l,n_1,n_2}^{\pm} = \mathbb{E}_{D,\boldsymbol{x}^{\pm}} \left[ \mathbb{E}_{\boldsymbol{c}} \left[ \psi(\boldsymbol{x} \cdot \hat{\boldsymbol{w}}(D,\boldsymbol{c})/\sqrt{N} + \hat{B}(D,\boldsymbol{c})) \right]^l \right]. \tag{46}$$

In addition, the predictive distribution does not fluctuate with respect to each realization of $D$ as assumed in Assumption 1. Therefore, we can obtain a nontrivial information about the distribution $\hat{s}^{\pm}(\boldsymbol{c}, D)$ if we obtain a nontrivial expression of $\Omega_{l,n_1,n_2}^{\pm}$ and continue the result to $n_1, n_2 \in \mathbb{R}$. In the following, we describe the handling of the expectation (43).

#### A.3.2 Handling of the expectation over the replicated system

Since the normalization constant $\Xi = \mathbb{E}_D \left[ \mathbb{E}_{\boldsymbol{c}}[Z(D,\boldsymbol{c})^{n_2}]^{n_1} \right]$ just yields the unity at $n_1, n_2 \to 0$, we focus on the numerator of the expectation over $\{\boldsymbol{\theta}_{\boldsymbol{a}}\} \sim \tilde{p}$ in $\Omega_{l,n_1,n_2}^{\pm}$. Instead of first taking the integral over $\{\boldsymbol{\theta}_{\boldsymbol{a}}\}$ as in the computation to obtain (44), we here first consider the average over $\boldsymbol{x}$ and $D$. Then, using independence of the each data point in $D$ and the reweighting coefficient $\boldsymbol{c}$, we obtain the following expression of $\Omega_{l,n_1,n_2}^{\pm}$:

$$\Omega_{l,n_1,n_2}^{\pm} \times \Xi = \int \mathbb{E}_{\boldsymbol{z}} \left[ \prod_{k=1}^{l} \psi(\pm \frac{1}{N} \boldsymbol{w}_{\boldsymbol{a}^{(k)}} \cdot \boldsymbol{v} + B_{\boldsymbol{a}^{(k)}} + \frac{1}{\sqrt{N}} \boldsymbol{w}_{\boldsymbol{a}^{(k)}} \cdot \boldsymbol{z}) \right]$$

$$\times\ \mathbb{E}_{\boldsymbol{z}^+,\{c_{a_1}^+\}}\left[\prod_{\boldsymbol{a}}e^{-\beta c_{a_1}^+ l^+(\frac{1}{N}\boldsymbol{w_a}\cdot\boldsymbol{v}+B_{\boldsymbol{a}}+\frac{1}{\sqrt{N}}\boldsymbol{w_a}\cdot\boldsymbol{z}^+)}\right]^{M^+}$$

$$\times\ \mathbb{E}_{\boldsymbol{z}^-,\{c_{a_1}^-\}}\left[\prod_{\boldsymbol{a}}e^{-\beta c_{a_1}^- l^-(-\frac{1}{N}\boldsymbol{w_a}\cdot\boldsymbol{v}+B_{\boldsymbol{a}}+\frac{1}{\sqrt{N}}\boldsymbol{w_a}\cdot\boldsymbol{z}^-)}\right]^{M^-}\prod_{i=1}^{N}\prod_{\boldsymbol{a}}e^{-\beta r_{\boldsymbol{w}_{\boldsymbol{a},i}}}d^{n_1 n_2}\boldsymbol{\theta}, \qquad (47)$$

where $c_{a_1}^+ \sim_{\text{iid}} p_c^+, c_{a_1}^- \sim_{\text{iid}} p_c^-, a_1 \in [n_1]$ and $z_i, z_i^\pm \sim_{\text{iid}} p_z, i \in [N]$. An important observation is that the noise $\boldsymbol{z}$ appears only through the form of the scaled inner product with $\boldsymbol{w_a}$. At $N \gg 1$, such quantity follows the Gaussian distribution for a fixed set of $\{\boldsymbol{w_a}\}$ due to the central limit theorem. Let us define $\tilde{\boldsymbol{u}} \in \mathbb{R}^l$ and $\boldsymbol{u}^\pm \in R^{n_1 \times n_2}$ as

$$\tilde{u}_k = \frac{1}{\sqrt{N}}\boldsymbol{w}_{\boldsymbol{a}^{(k)}} \cdot \boldsymbol{z}, \qquad (48)$$

$$u_{\boldsymbol{a}}^+ = \frac{1}{\sqrt{N}}\boldsymbol{w_a} \cdot \boldsymbol{z}^+, \qquad (49)$$

$$u_{\boldsymbol{a}}^- = \frac{1}{\sqrt{N}}\boldsymbol{w_a} \cdot \boldsymbol{z}^-. \qquad (50)$$

Then, these quantities follows the Gaussian distributions as $\tilde{\boldsymbol{u}} \sim \mathcal{N}(0, \tilde{S}), \boldsymbol{u}^\pm \sim \mathcal{N}(0, S)$, where the covariance matrices are given as

$$\tilde{S}_{k,k'} = \Delta\frac{1}{N}\boldsymbol{w}_{\boldsymbol{a}^{(k)}} \cdot \boldsymbol{w}_{\boldsymbol{a}^{(k')}}, \qquad (51)$$

$$S_{\boldsymbol{a},\boldsymbol{b}} = \Delta\frac{1}{N}\boldsymbol{w_a} \cdot \boldsymbol{w_b}. \qquad (52)$$

Also, the center of the clusters $\boldsymbol{v}$ only appears through the inner product $\boldsymbol{w_a} \cdot \boldsymbol{v}/N$. Thus, the integrand in (47) depends on $\{\boldsymbol{w_a}\}$ only through their inner products, which capture the geometric relations between the estimators and the centroid of clusters $\boldsymbol{v}$. We call these quantities as *order parameters*.

The above observation indicates that the factors regarding the loss function in (47) can be evaluated by Gaussian integrals once the order parameters are fixed. To implement this idea, we insert the trivial identities of the delta functions

$$1 = \prod_{\boldsymbol{a} \neq \boldsymbol{b}}\int \delta_{\text{d}}(NQ_{\boldsymbol{ab}} - \boldsymbol{w_a} \cdot \boldsymbol{w_b})dQ_{\boldsymbol{ab}}, \qquad (53)$$

$$1 = \prod_{\boldsymbol{a}}\int \delta_{\text{d}}(Nm_{\boldsymbol{a}} - \boldsymbol{w_a} \cdot \boldsymbol{v})dm_{\boldsymbol{a}}, \qquad (54)$$

into (47). Then $\Omega_{l,n_1,n_2}^\pm \times \Xi$ can be rewritten as follows

$$\Omega_{l,n_1,n_2}^\pm \times \Xi = \int \mathbb{E}_{\tilde{\boldsymbol{u}}\sim\mathcal{N}(0,\tilde{S})}\left[\prod_{k=1}^{l}\psi(\pm m_{\boldsymbol{a}^{(k)}} + B_{\boldsymbol{a}^{(k)}} + \tilde{u}_k)\right]e^{N\alpha^+\varphi^+ + N\alpha^-\varphi^-}$$

$$\times \prod_{\boldsymbol{a}\neq\boldsymbol{b}}\delta_{\text{d}}(NQ_{\boldsymbol{ab}} - \boldsymbol{w_a} \cdot \boldsymbol{w_b})\prod_{\boldsymbol{a}}\delta_{\text{d}}(Nm_{\boldsymbol{a}} - \boldsymbol{w_a} \cdot \boldsymbol{v})\prod_{i=1}^{N}\prod_{\boldsymbol{a}}e^{-\beta r_{\boldsymbol{w}_{\boldsymbol{a},i}}}d^{n_1 n_2}\boldsymbol{\theta}d\Theta, \qquad (55)$$

$$\varphi^+ = \log\mathbb{E}_{\boldsymbol{u}^+\sim\mathcal{N}(0,S),\{c_{a_1}^+\}}\left[\prod_{\boldsymbol{a}}e^{-\beta c_{a_1}^+ l^+(m_{\boldsymbol{a}}+B_{\boldsymbol{a}}+u_{\boldsymbol{a}}^+)}\right], \qquad (56)$$

$$\varphi^- = \log\mathbb{E}_{\boldsymbol{u}^-\sim\mathcal{N}(0,S),\{c_{a_1}^-\}}\left[\prod_{\boldsymbol{a}}e^{-\beta c_{a_1}^- l^-(-m_{\boldsymbol{a}}+B_{\boldsymbol{a}}+u_{\boldsymbol{a}}^-)}\right] \qquad (57)$$

where $\Theta$ is the collection of the variables $\{Q_{\boldsymbol{ab}}\}, \{m_{\boldsymbol{a}}\}$. To complete the remaining integrals over $\{\boldsymbol{w_a}\}$, we use the Fourier representations of the delta functions:

$$\delta_{\mathrm{d}}(NQ_{\boldsymbol{ab}} - \boldsymbol{w_a} \cdot \boldsymbol{w_b}) = \frac{1}{2\pi} \int e^{(NQ_{\boldsymbol{ab}} - \boldsymbol{w_a} \cdot \boldsymbol{w_b})\tilde{Q}_{\boldsymbol{ab}}} d\tilde{Q}_{\boldsymbol{ab}}, \tag{58}$$

$$\delta_{\mathrm{d}}(Nm_{\boldsymbol{a}} - \boldsymbol{w_a} \cdot \boldsymbol{v}) = \frac{1}{2\pi} \int e^{-(Nm_{\boldsymbol{a}} - \boldsymbol{w} \cdot \boldsymbol{v})\tilde{m}_{\boldsymbol{a}}} d\tilde{m}_{\boldsymbol{a}}. \tag{59}$$

After using these expressions, the integrals over $\{w_{\boldsymbol{a},i}\}$ can be independently performed for each $i$, leading to the following expression of $\Omega^{\pm}_{l,n_1,n_2} \times \Xi$:

$$\Omega^{\pm}_{l,n_1,n_2} \times \Xi = \int \mathbb{E}_{\tilde{\boldsymbol{u}} \sim \mathcal{N}(0,\tilde{S})} \left[ \prod_{k=1}^{l} \psi(\pm m_{\boldsymbol{a}^{(k)}} + B_{\boldsymbol{a}^{(k)}} + \tilde{u}_k) \right] e^{N\mathcal{G}(\Theta,\hat{\Theta})} d\Theta d\hat{\Theta}, \tag{60}$$

$$\mathcal{G}(\Theta,\hat{\Theta}) = \frac{1}{2} \sum_{\boldsymbol{a},\boldsymbol{b}} Q_{\boldsymbol{ab}} \tilde{Q}_{\boldsymbol{ab}} - \sum_{\boldsymbol{a}} m_{\boldsymbol{a}} \tilde{m}_{\boldsymbol{a}} + \alpha^+ \varphi^+ + \alpha^- \varphi^- + \varphi_w, \tag{61}$$

$$\varphi_w = \log \int e^{-\frac{1}{2} \sum_{\boldsymbol{a},\boldsymbol{b}} \tilde{Q}_{\boldsymbol{ab}} w_{\boldsymbol{a}} w_{\boldsymbol{b}} + \sum_{\boldsymbol{a}} \tilde{m}_{\boldsymbol{a}} w_{\boldsymbol{a}}} \prod_{\boldsymbol{a}} e^{-\beta r(w_{\boldsymbol{a}})} d^{n_1 n_2} w, \tag{62}$$

where $\hat{\Theta}$ is the correction of variables $\{\tilde{Q}_{\boldsymbol{ab}}\}, \{\tilde{m}_{\boldsymbol{a}}\}$. At $N \gg 1$, using the saddle point method, it is understood that the integral over $\Theta, \hat{\Theta}$ is dominated at the extremum $\mathrm{extr}_{\Theta,\hat{\Theta}} \mathcal{G}(\Theta,\hat{\Theta})$. Hence, we obtain

$$\Omega^{\pm}_{l,n_1,n_2} \times \Xi = \mathbb{E}_{\tilde{\boldsymbol{u}} \sim \mathcal{N}(0,\tilde{S})} \left[ \prod_{k=1}^{l} \psi(\pm m_{\boldsymbol{a}^{(k)}} + B_{\boldsymbol{a}^{(k)}} + \tilde{u}_k) \right] e^{N\mathcal{G}(\Theta,\hat{\Theta})}, \tag{63}$$

where $\Theta, \hat{\Theta}$ is obtained by the extremum condition $\mathrm{mathop}_{\Theta,\hat{\Theta}} \mathcal{G}(\Theta,\hat{\Theta})$.

At this point, we can obtain the general expression of the extreme condition. Let us define densities $\tilde{p}_{\mathrm{eff},w}$ and $\tilde{p}^{\pm}_{\mathrm{eff},s}$ as follows:

$$\tilde{p}_{\mathrm{eff},w}(\boldsymbol{w}) = \frac{1}{Z_{\mathrm{eff},w}} e^{-\frac{1}{2} \sum_{\boldsymbol{a},\boldsymbol{b}} \tilde{Q}_{\boldsymbol{ab}} w_{\boldsymbol{a}} w_{\boldsymbol{b}} + \sum_{\boldsymbol{a}} \tilde{m}_{\boldsymbol{a}} w_{\boldsymbol{a}}} \prod_{\boldsymbol{a}} e^{-\beta r(w_{\boldsymbol{a}})}, \tag{64}$$

$$\tilde{p}^{\pm}_{\mathrm{eff},u}(\boldsymbol{u}|\{c^{\pm}_{a_1}\}) = \frac{1}{Z^{\pm}_{\mathrm{eff},u}} \prod_{\boldsymbol{a}} e^{-\beta c^{\pm}_{a_1} l^{\pm}(\pm m_{\boldsymbol{a}} + B_{\boldsymbol{a}} + u_{\boldsymbol{a}})} \times \mathcal{N}(\boldsymbol{u}; \boldsymbol{0}, \Delta Q), \tag{65}$$

where $Q = [Q_{\boldsymbol{ab}}] \in \mathbb{R}^{n_1 n_2 \times n_1 n_2}$. Then, the exremum condition is given as follows:

$$Q_{\boldsymbol{ab}} = \mathbb{E}_{\boldsymbol{w} \sim p_{\mathrm{eff},w}} [w_{\boldsymbol{a}} w_{\boldsymbol{b}}], \tag{66}$$

$$m_{\boldsymbol{a}} = \mathbb{E}_{\boldsymbol{w} \sim p_{\mathrm{eff},w}} [w_{\boldsymbol{a}}], \tag{67}$$

$$\begin{aligned}
\tilde{Q}_{\boldsymbol{ab}} = &-\alpha^+ \Delta \mathbb{E}_{\{c^+_{a_1}\}} \Big[ \mathbb{E}_{\boldsymbol{u}^+ \sim p^+_{\mathrm{eff},u}} \big[ c^+_{a_1} c^+_{b_1} \beta^2 \partial l^+(m_{\boldsymbol{a}} + B_{\boldsymbol{a}} + u_{\boldsymbol{a}}) \partial l^+(m_{\boldsymbol{b}} + B_{\boldsymbol{b}} + u_{\boldsymbol{b}}) \\
&- \beta c^+_{a_1} \partial^2 l^+(m_{\boldsymbol{a}} + B_{\boldsymbol{a}} + u_{\boldsymbol{a}}) \mathbb{1}(\boldsymbol{a} = \boldsymbol{b}) \big] \Big] \\
&- \alpha^- \Delta \mathbb{E}_{\{c^-_{a_1}\}} \Big[ \mathbb{E}_{\boldsymbol{u}^- \sim p^-_{\mathrm{eff},u}} \big[ c^-_{a_1} c^-_{b_1} \beta^2 \partial l^-(-m_{\boldsymbol{a}} + B_{\boldsymbol{a}} + u_{\boldsymbol{a}}) \partial l^-(-m_{\boldsymbol{b}} + B_{\boldsymbol{b}} + u_{\boldsymbol{b}}) \\
&- \beta c^-_{a_1} \partial^2 l^-(-m_{\boldsymbol{a}} + B_{\boldsymbol{a}} + u_{\boldsymbol{a}}) \mathbb{1}(\boldsymbol{a} = \boldsymbol{b}) \big] \Big],
\end{aligned} \tag{68}$$

$$\begin{aligned}
\tilde{m}_{\boldsymbol{a}} = &-\alpha^+ \mathbb{E}_{\{c^+_{a_1}\}} \Big[ \mathbb{E}_{\boldsymbol{u}^+ \sim p^+_{\mathrm{eff},u}} \big[ \beta c^+_{a_1} \partial l^+(m_{\boldsymbol{a}} + B_{\boldsymbol{a}} + u_{\boldsymbol{a}}) \big] \Big] \\
&+ \alpha^- \mathbb{E}_{\{c^-_{a_1}\}} \Big[ \mathbb{E}_{\boldsymbol{u}^- \sim p^-_{\mathrm{eff},u}} \big[ \beta c^-_{a_1} \partial l^-(-m_{\boldsymbol{a}} + B_{\boldsymbol{a}} + u_{\boldsymbol{a}}) \big] \Big].
\end{aligned} \tag{69}$$

Also, the next is added if the bias is estimated:

$$\begin{aligned}
0 = &\alpha^+ \mathbb{E}_{\{c^+_{a_1}\}} \Big[ \mathbb{E}_{\boldsymbol{u}^+ \sim p^+_{\mathrm{eff},u}} \big[ c^+_{a_1} \partial l^+(m_{\boldsymbol{a}} + B_{\boldsymbol{a}} + u_{\boldsymbol{a}}) \big] \Big] \\
&+ \alpha^- \mathbb{E}_{\{c^-_{a_1}\}} \Big[ \mathbb{E}_{\boldsymbol{u}^- \sim p^-_{\mathrm{eff},u}} \big[ c^-_{a_1} \partial l^-(-m_{\boldsymbol{a}} + B_{\boldsymbol{a}} + u_{\boldsymbol{a}}) \big] \Big].
\end{aligned} \tag{70}$$

### A.3.3 Replica symmetric ansatz

Since $\mathcal{G}(\Theta, \hat{\Theta})$ depends on the discrete nature of $n_1, n_2$, we cannot analytically continue the result to $n_1, n_2 \in \mathbb{R}$. To make an analytical continuation, the key issue is to identify the correct form of the extreme condition. The choice that reflect the symmetry of the replicated system when $n_1, n_2 \in \mathbb{N}$ is the following:

$$Q_{\boldsymbol{ab}} = q + \mathbb{1}(a_1 = b_1)v + \mathbb{1}(\boldsymbol{a} = \boldsymbol{b})\frac{\chi}{\beta}, \tag{71}$$

$$m_{\boldsymbol{a}} = m, \tag{72}$$

$$\tilde{Q}_{\boldsymbol{ab}} = -\beta^2\hat{\chi} - \mathbb{1}(a_1 = b_1)\beta^2\hat{v} + \mathbb{1}(\boldsymbol{a} = \boldsymbol{b})\beta\hat{Q}, \tag{73}$$

$$\tilde{m}_{\boldsymbol{a}} = \beta\hat{m}, \tag{74}$$

$$B_{\boldsymbol{a}} = B. \tag{75}$$

Evaluating the replicated system using this symmetric form of the extremum and analytically continuing as $n_1, n_2 \to 0$ is called the replica symmetric (RS) ansatz. When considering the estimation of the linear models with convex loss, it is empirically known that the RS ansatz yields the exact result (Obuchi & Kabashima, 2019; Takahashi & Kabashima, 2020).

Using this symmetric form of the extremum, after simple algebra, one can find that $\mathcal{G}$ can be analytically continued to $n_1, n_2 \in \mathbb{R}$ and also $\mathcal{G} = \mathcal{O}(n_1) + \mathcal{O}(n_2)$. Furthermore, the remaining factor $\mathbb{E}_{\tilde{\boldsymbol{u}}\sim\mathcal{N}(0,\tilde{S})}\left[\prod_{k=1}^{l}\psi(\pm m_{\boldsymbol{a}^{(k)}} + B_{\boldsymbol{a}^{(k)}} + \tilde{u}_k)\right]$ can be rewritten as follows

$$\mathbb{E}_{\tilde{\boldsymbol{u}}\sim\mathcal{N}(0,\tilde{S})}\left[\prod_{k=1}^{l}\psi(\pm m_{\boldsymbol{a}^{(k)}} + B_{\boldsymbol{a}^{(k)}} + \tilde{u}_k)\right] = \mathbb{E}_{\xi_u\sim\mathcal{N}(0,\Delta)}\left[\mathbb{E}_{\eta_u\sim\mathcal{N}(0,\Delta)}\left[\psi(\pm m + B + \sqrt{q}\xi_u + \sqrt{v}\eta_u)\right]^l\right]. \tag{76}$$

After all, we obtain the following result:

$$\lim_{\beta\to\infty}\lim_{n_1,n_2\to 0}\Omega_{l,n_1,n_2}^{\pm} = \mathbb{E}_{\xi_u\sim\mathcal{N}(0,\Delta)}\left[\mathbb{E}_{\eta_u\sim\mathcal{N}(0,\Delta)}\left[\psi(\pm m + B + \sqrt{q}\xi_u + \sqrt{v}\eta_u)\right]^l\right], \tag{77}$$

where $m, B, q, v$ are determined by the extreme condition under the RS ansatz at $\beta \to \infty$, which is given as the self-consistent equation in Definition 1. This yields the Claim 2.

Similarly, by considering the expectations such as $\mathbb{E}_{\{\boldsymbol{\theta}_a\}\sim\tilde{p}}[\boldsymbol{w}_{\boldsymbol{a}}^{\top}\boldsymbol{w}_{\boldsymbol{b}}/N], \mathbb{E}_{\{\boldsymbol{\theta}_a\}\sim\tilde{p}}[\boldsymbol{w}_{\boldsymbol{a}}^{\top}\boldsymbol{v}/N], \mathbb{E}_{\{\boldsymbol{\theta}_a\}\sim\tilde{p}}[B_{\boldsymbol{a}}]$, one can obtain Claim 1.

## B  Cross-checking with numerical experiments at small $N$

In the main text we observed that the empirical distribution of the weight vector is almost perfectly consistent with the theory when $N = 2^{13}$. In this appendix, we also show that the experimental results can be well approximated by the theoretical predictions even for a smaller size with $N = 2^{10}$ as shown in Figure 12. Note that the average number of positive examples $M^+ \simeq 5$ when $M^+/M = 0.01, N = 2^{10}, (M^+ + M^-)/N = 1/2$, which is obviously far from the LSL. Therefore, this result may indicate a rapid convergence to the LSL.

## C  On US estimator without replacement

In this appendix, we show that the $F$-measure for the estimator obtained by US when using sampling without replacement does not depends on the size of the excess majority samples.

The $F$-measure for the estimator obtained by US, which corresponds to the case of $K = 1$ in Claim 4, is determined by $\bar{q} = q + v$. Let us also define $\hat{\bar{\chi}} = \hat{\chi} + \hat{v}$. Then, by adding both side of equations (20) and (22) as well as (24) and (26), and taking the average of $c^{\pm} \sim p_c^{\pm}$ explicitly, the following modified self-consistent equations that determine $\bar{q}, m, B$ are obtained as follows.

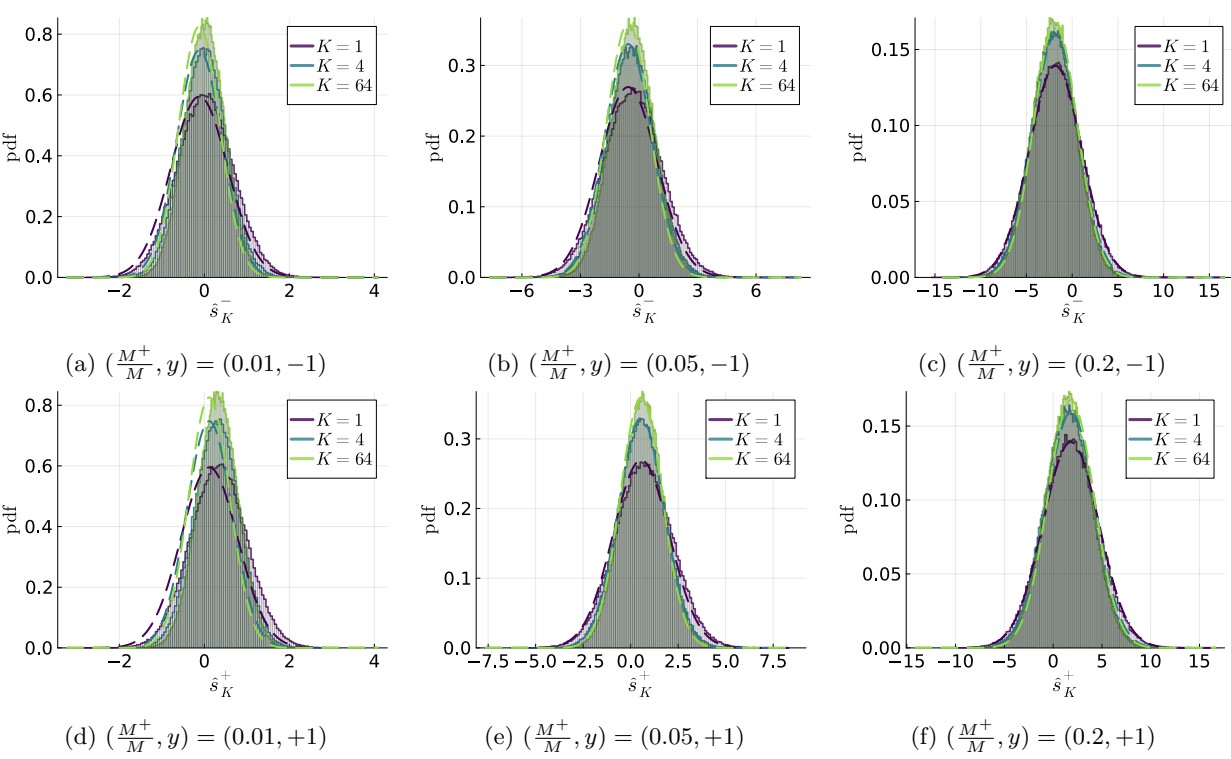

Figure 12: Comparison between the empirical distribution of $\hat{s}_K^\pm$ (histogram), which is obtained by a single realization of the training data $D$ of finite size with $N = 2^{10}$, and the theoretical prediction in Claim 3 (dashed). Different colors represent resampling averages with different numbers of realizations of reweighting coefficient **c**.

**Claim 5 (Modified self-consistent equations for US)** *Let $\bar{q} = q + v$ and $\hat{\bar{\chi}} = \hat{\chi} + \hat{v}$. Then, the set of quantities $\bar{q}, m, \chi$ and $\hat{Q}, \hat{\bar{\chi}}, \hat{m}$ are given as the solution of the following set of modified self-consistent equations when subsampling (8)-(9) is used for the resampling method. Let $\bar{w}$ and $\bar{u}^\pm$ be the solution of the following one-dimensional randomized optimization problems:*

$$\bar{w} = \underset{w \in \mathbb{R}}{\arg\min} \frac{\hat{Q}}{2}w^2 - \bar{h}_w w + \lambda r(w), \tag{78}$$

$$\bar{u}^\pm = \underset{u \in \mathbb{R}}{\arg\min} \frac{u^2}{2\chi\Delta} + l^\pm(u + \bar{h}_u^\pm), \tag{79}$$

*where*

$$\bar{h}_w = \hat{m} + \sqrt{\hat{\bar{\chi}}}\xi_w, \quad \xi_w, \sim \mathcal{N}(0,1), \tag{80}$$

$$\bar{h}_u^\pm = B \pm m + \sqrt{\bar{q}}\xi_u, \quad \xi_u \sim \mathcal{N}(0,\Delta). \tag{81}$$

*Then, the modified self-consistent equations are given as follows:*

$$\chi = E_{\xi_w, \sim \mathcal{N}(0,1)}\left[\frac{d}{d\bar{h}_w}\bar{w}\right], \tag{82}$$

$$\bar{q} = \mathbb{E}_{\xi_w \sim \mathcal{N}(0,1)}\left[\bar{w}^2\right], \tag{83}$$

$$m = E_{\xi_w \sim \mathcal{N}(0,1)}\left[\bar{w}\right], \tag{84}$$

$$\hat{Q} = -\frac{\alpha^+}{\chi}\mathbb{E}_{\xi_u \sim \mathcal{N}(0,\Delta)}\left[\frac{d}{d\bar{h}_u^+}\bar{u}^+ + \frac{d}{d\bar{h}_u^-}\bar{u}^-\right], \tag{85}$$

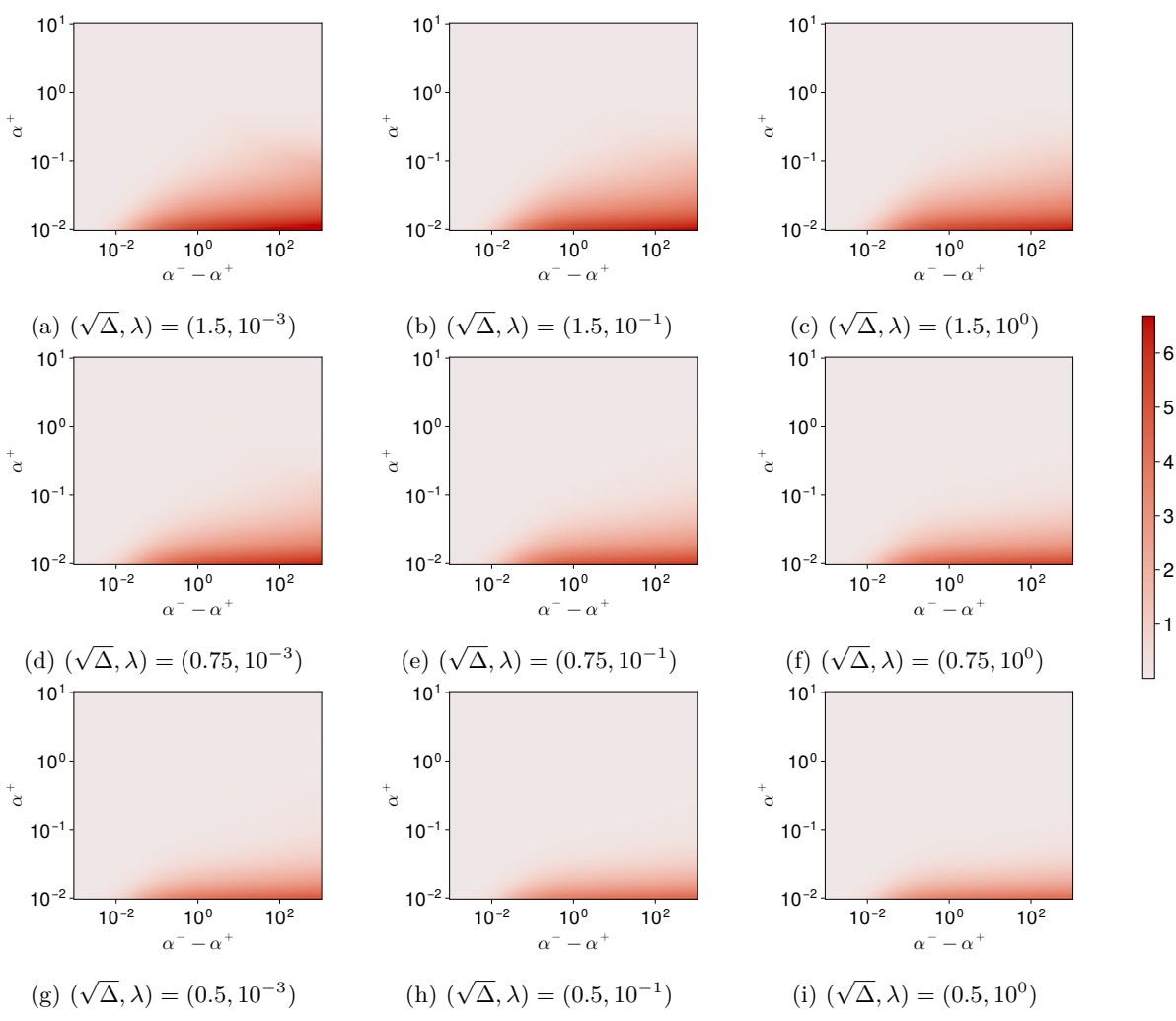

Figure 13: The heatmap plot for the relative $F$-measure in log-scale: $\log_{10} \mathcal{F}_{\text{UB}}/\mathcal{F}_{\text{weighting}}$, where $\mathcal{F}_{\text{UB}}$ and $\mathcal{F}_{\text{weighting}}$ are the $F$-measures for UB and the weighting method with naive coefficients, respectively.

$$\hat{\hat{\chi}} = \frac{\alpha^+}{\Delta\chi^2}\mathbb{E}_{\xi_u \sim \mathcal{N}(0,\Delta)}\left[(\bar{\mathsf{u}}^+)^2 + (\bar{\mathsf{u}}^-)^2\right], \tag{86}$$

$$\hat{m} = \frac{\alpha^+}{\Delta\chi}\mathbb{E}_{\xi_u, \sim \mathcal{N}(0,\Delta)}\left[\bar{\mathsf{u}}^+ - \bar{\mathsf{u}}^-\right]. \tag{87}$$

*Furthermore, depending on whether the bias is estimated or fixed, the following equation is added to the modified self-consistent equation*

$$\begin{cases} 0 = \mathbb{E}_{\xi_u \sim \mathcal{N}(0,\Delta)}\left[\bar{\mathsf{u}}^+ + \bar{\mathsf{u}}^-\right], & \text{when bias is estimated} \\ B = \bar{B}, & \text{when bias is fixed} \end{cases}. \tag{88}$$

It is clear that the above modified self-consistent equation only depends on $\alpha^+$ and does not depends on $\alpha^- - \alpha^+$. Therefore, the $F$-measure of US cannot be improved by the excess examples the majority class[4].

---

[4]This is exactly the same with the equations to determine the behavior of linear classifiers trained on the data generated from symmetric Gaussian mixture model with a data size $\alpha^+ N$ and an input dimension $N$ obtained in (Mignacco et al., 2020).

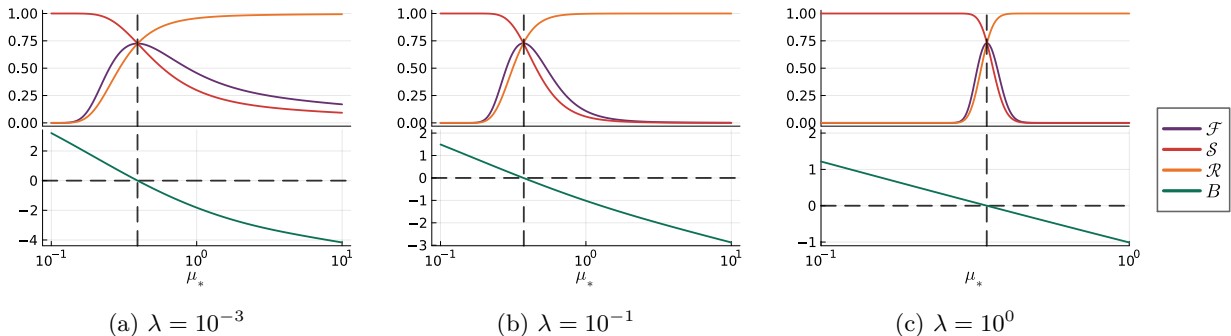

(a) $\lambda = 10^{-3}$        (b) $\lambda = 10^{-1}$        (c) $\lambda = 10^{0}$

Figure 14: The resampling rate dependence of the $F$-measure when using sampling with replacement for selected values of $\lambda$. Here the parameters are set as $(\alpha^+, \alpha^-, \sqrt{\Delta}) = (0.05, 0.1, 0.75)$. The vertical line shows the value of the resampling rate that makes the bias term to be zero.

## D SW with naive re-weighting coefficient

Figure 13 show the heatmap plot for the relative $F$-measure in log-scale: $\log_{10} \mathcal{F}_{\text{UB}}/\mathcal{F}_{\text{weighting}}$, where $\mathcal{F}_{\text{UB}}$ and $\mathcal{F}_{\text{weighting}}$ are the $F$-measures for UB and the weighting method with the naive coefficients $\gamma_{\text{naive}}^{\pm} = (M^+ + M^-)/(2M^{\pm})$. It is clear that, with the naive reweighting coefficient, the $F$-measure of SW is catastrophically worse that of UB, especially when the minority class data is small and the excess majority samples are large. This suggests that delicate tuning of the reweighting coefficients are required in overparameterized settings.

## E Resampling rate dependence for sampling with replacement

In this section, we show the resampling rate dependence of the $F$-measure when using sampling with replacement. As shown in Figure 14, the $F$-measure takes the highest value when the bias term is estimated to be zero. This is because the $F$-measure considered here treats the positive and negative examples equally.

