# OpenReview forum: "A replica analysis of under-bagging"
_TMLR — Accepted by TMLR_

### Review · Reviewer_eAxG · 2024-05-11

**Summary Of Contributions:**

The submitted paper presents an asymptotic analysis of the under-bagging (UB) method for the training of classifiers in the case of unbalanced datasets. The authors derive a collection of self-consistent equations for a set of low-dimensional order parameters, expressing the variance of the estimator and its alignment with the signal, in the asymptotic proportional regime, in which the dimension of the data points and size of the dataset diverge keeping a fix ration. By means of such order parameters, the $F$-score of the estimator is computed. The same analysis extends to the case of estimators obtained via under-sampling (US) and simple reweighting (SW), allowing for a direct performance comparison of the various methods. As a result, it is shown that the ensembling procedure within the under-bagging strategy is beneficial, as it leads to a suppression of the sampling fluctuations.

**Audience:**

Yes

**Broader Impact Concerns:**

No broader impact statement is required for this paper.

**Claims And Evidence:**

Yes

**Requested Changes:**

I list below some remarks on the paper.
- As a general recommendation, I advise the authors to add an explicit definition of the UB estimator. Surprisingly (considering the fact that it is central in the paper), the UB setting is compared with, e.g., the US and SW but is not described in detail. I assume that the UB estimator considered by the authors is obtained by ensembling an infinite number ($K\to+\infty$) of under-sampled set-ups. If this is the case, this has to be stated explicitly somewhere in the manuscript. For example, at the end of page 8, the authors specify that ```UB with $K>1$ is [has?] a higher value of the $F$-measure``` which suggests that the authors consider UB any $K>1$. However, in Section 4, $K$ is not discussed anymore: the results and the analogy with ensembling results by [D'Ascoli et al.](https://proceedings.mlr.press/v119/d-ascoli20a.html) or [Loureiro et al.](https://arxiv.org/abs/2201.13383v1) leads to infer that therein $K\to+\infty$: if this is the case, this should be stated explicitly.
- Unless I missed it, it is always assumed, but never specified, that the true distribution of the data points is balanced between the two clusters. This should be stated somewhere, for example in the ```Problem setup``` section.
- The standard definition of the [$F$-score]() $\mathcal F$ in Eq 36 involves the *precision* in place of the *specificity*: I suppose that the authors followed [Wallace et al.](https://ieeexplore-ieee-org.ezproxy.unibo.it/document/6137280) who used specificity to avoid dependence on the prevalence. I think that this should be specified explicitly.
- It is expected that the subset of figure for $N=2^{10}$ in Fig. 1 are indeed affected by strong finite-size effects (it is maybe surprising that they are not that strong) as in this case $\frac{M^+}{M}=10^{-2}$ means that $M^+\sim 5$. I wonder if it is sensible to keep them in the text or not given that the main interest is the LSL.
- As observed by the authors themselves, Claim 5 coincides with the results of [Mignacco et al.](https://proceedings.mlr.press/v119/mignacco20a.html) as, in the US case, the problem is intrinsically a classification task of $\alpha^+N$ points. Is the conclusion given at the end of Sec. 4.1 an immediate consequence of the fact that, in this case, $\alpha^-$ plays no role? If this is the case, the authors might consider moving it to the Appendix.

Finally, I list here some typos that the authors might fix. I think $p_c^+(c^-)$ in Eq. 7 should be $p_c^-(c^-)$. Also in Section 3.1 $N$ is referred to as *system size*, although it is the dimension. In Eqs 42 and 44, $E$ should be replaced by $\mathbb E$. The label of Fig 2 starts with ```Comparison of the comparison```. After Eq 72 there is a reference to a ```mathop``` which is likely meant to be $\min$ (as a general remark, the Appendix appears to be in a very sketchy form).

**Strengths And Weaknesses:**

*Strenghts*
The paper quantifies the beneficial effects of bagging by relating the improved performances to the suppression of sampling fluctuations, at the same time exploiting in its entirety the information contained in the abundance of the dominant class in a classification task. The paper complements some recent findings in a similar setting, eg the paper by [Clarté et al.](https://arxiv.org/pdf/2402.13622) on bootstrap and subsampling in high dimensional regression.

*Weaknesses*
The paper appears as a variation, in a new setting, of methodologically similar investigations available in the literature, in which the replica method is used to extract asymptotic performances of convex GLM under ensembling and bootstrap. The replica method is non-rigorous, but it proved to be a very powerful technique, and, nevertheless, the theoretical results are supported by numerical simulations. Bayes-optimal references for the performances are not discussed.

---

> ### Author Response · Authors · 2024-06-11
> **Response to reviewer eAxG (1/1)**
>
> Thank you for your constructive reviews. We appreciate your consideration to improve the work. Below are the responses to each comment.
> > As a general recommendation, I advise the authors to add an explicit definition of the UB estimator. Surprisingly (considering the fact that it is central in the paper), the UB setting is compared with, e.g., the US and SW but is not described in detail. I assume that the UB estimator considered by the authors is obtained by ensembling an infinite number ($K\to+\infty$) of under-sampled set-ups. If this is the case, this has to be stated explicitly somewhere in the manuscript. For example, at the end of page 8, the authors specify that UB with $K>1$ is [has?] a higher value of the $F$-measure which suggests that the authors consider UB any $K>1$. However, in Section 4, $K$ is not discussed anymore: the results and the analogy with ensembling results by D'Ascoli et al. or Loureiro et al. leads to infer that therein $K\to+\infty$: if this is the case, this should be stated explicitly.
>
> We admit that the definition of the UB was vague in the previous manuscript. As you guessed, $K\to+\infty$ is used in Section 4, and finite $K$ case is used only for the sanity checks in Section 3. This point is now explicitly explained at the beginning of Section 4.
>
> > "Unless I missed it, it is always assumed, but never specified, that the true distribution of the data points is balanced between the two clusters. This should be stated somewhere, for example in the Problem setup section.
>
> > The standard definition of the $F$-score $\mathcal F$ in Eq 36 involves the precision in place of the specificity: I suppose that the authors followed Wallace et al. who used specificity to avoid dependence on the prevalence. I think that this should be specified explicitly.
>
> As pointed out, the text-book definition of the $F$-measure involves the precision instead of the specificity. In the classification problem, we cannot uniquely determine the performance metric unless we specify how much importance is given to errors for each class. To make the matters simple, in this paper we have taken the position that errors are treated equally for positive and negative exampled data, and have used specificity instead of precision, although we do understand that this metric is not the unique choice. Then, the modified $F$-measure is independent of the prevalence or the imbalance of the test data, which simplifies the discussion. We have clarified this point in Section 2.
>
> > It is expected that the subset of figure for $N=2^{10}$ in Fig. 1 are indeed affected by strong finite-size effects (it is maybe surprising that they are not that strong) as in this case $\frac{M^+}{M}=10^{-2}$ means that $M^+\sim 5$. I wonder if it is sensible to keep them in the text or not given that the main interest is the LSL.
>
> Thank you for the constructive advise.  It is true that  $2^{10}$ is very far from LSL, and it was also pointed out by ihfu that there are too many panels which makes the figure difficult to read. Thus, we have moved the $2^{10}$ cases to Appendix B.
>
> > As observed by the authors themselves, Claim 5 coincides with the results of Mignacco et al. as, in the US case, the problem is intrinsically a classification task of $\alpha^+N$ points. Is the conclusion given at the end of Sec. 4.1 an immediate consequence of the fact that, in this case, $\alpha^-$ plays no role? If this is the case, the authors might consider moving it to the Appendix.
>
> As pointed out, this is the immediate consequence of the fact that $\alpha^- - \alpha^+$  plays no role. Thus, we have moved it to Appendix C as suggested.
>
> > Finally, I list here some typos that the authors might fix. I think $p_c^+(c^-)$ in Eq. 7 should be $p_c^-(c^-)$. Also in Section 3.1 $N$ is referred to as system size, although it is the dimension. In Eqs 42 and 44, $E$ should be replaced by $\mathbb E$. The label of Fig 2 starts with Comparison of the comparison. After Eq 72 there is a reference to a mathop which is likely meant to be $\min$ (as a general remark, the Appendix appears to be in a very sketchy form).
>
> Thank you for your careful reading. We have fixed all of them.

---

### Review · Reviewer_ihfu · 2024-06-03

**Summary Of Contributions:**

This paper studies the performance of different statistical methods for linear classification when the training data is imbalanced between two classes (i.e one class is more represented in the training data than the other). The methods considered are undersampling (US), underbagging (UB) which combines undersampling and ensembling, and sample weighting (SW). These methods are studied in the asymptotic regime where the dimension and number of data are large and of same order, and in a synthetic data model (Gaussian mixture model).
The author shows that the performance for the F-score of UB is improved by having more data of the majority class, while US is insensitive to the imbalance. On the other hand, the performance of SW decreases with the data imbalance, hinting that this method is not efficient to counter data imbalance.
On the technical side, the author use the standard replica method to compute the asymptotic properties of the different methods.

**Audience:**

Yes

**Broader Impact Concerns:**

Data imbalance is an important topic in statistics. However I do not think that this paper requires adding a Broader Impact Statement as it is concerned with a theoretical analysis of existing methods, and the analysis is done on synthetic data.

**Claims And Evidence:**

Yes

**Requested Changes:**

I request no change that would be critical for acceptance, but the following changes would strenghten the work in my opinion :
* Reorganizing the main so that the state-evolution equation are more easily understandable for readers unfamiliar with these.
* Improving the readability of the figures

**Strengths And Weaknesses:**

1) **Strengths** :
- The contributions of the paper and its conclusion on the performance of the methods are clear, its conclusions are interesting and might be of practical relevance. Also, the authors provide numerous plots showing the sharpness of their predictions in the large system limit (LSL). The appendix containing the replica computation is overall clear.

2) **Weaknesses** :
- In general, I think the paper would be difficult to understand for a reader not familiar with the replica computation / state-evolution equations. Indeed, the structure of the argument might be weird for an unfamiliar reader : the author first introduces the state-evolution equations without detailing the interpretation of the overlaps $m, q, v$, making the equations . Only after (Claim 1) do they provide this interpretation. I think it would be helpful
- The analysis of the method is made only at constant $\lambda$ and not at $\lambda$ that would be cross-validated to minimize a validation loss, as it is done in practice. It would be interesting to see how choosing $\lambda$ "optimally" changes the results of the paper.

3) **Other remarks** :
- Formulating Assumption 1 as an assumption and not as a property of the problem seems a bit strange to me. Indeed, it is standard that the macroscopic properties do not fluctuate with $D$ in the LSL.
- "estiamted" Equation 29
- “Comparison of the comparison of” in Figure 2 might be a typo ?
- Figure 1 : not very readable because of the many plots, especially using the two different sizes $N = 2^{10}$ / $2^{13}$ feel redundant
- In general, the presentation of the Figures could be improved to be more readable. For instance, the legend may not be required in all subplots for Figure 2 and clusters the figure.
- "mathop" is not defined (maybe the author meant "ext" instead ?) just after equation 72

---

> ### Author Response · Authors · 2024-06-11
> **Response to reviewer ihfu (1/1)**
>
> Thank you for your careful reading and reviews. We appreciate your consideration to improve the work. Below are the responses to each comment.
>
> # Weakness
> > In general, I think the paper would be difficult to understand for a reader not familiar with the replica computation / state-evolution equations. Indeed, the structure of the argument might be weird for an unfamiliar reader : the author first introduces the state-evolution equations without detailing the interpretation of the overlaps $m, q, v$, making the equations . Only after (Claim 1) do they provide this interpretation. I think it would be helpful
>
> We admit that it may be difficult to read Definition 1 that shows the state-evolution (SE) equations. However, since all of the following claims are based on the solution of the SE equations and the interpretation of $\Theta$,  it seems unavoidable to show Definition 1and Claim 1 first.  For readers unfamiliar with replica analysis, we added a guide on reading order. We hope this will improve the readability to some extent.
>
> > The analysis of the method is made only at constant $\lambda$ and not at $\lambda$ that would be cross-validated to minimize a validation loss, as it is done in practice. It would be interesting to see how choosing $\lambda$ "optimally" changes the results of the paper.
>
> Thank you for raising this point. As pointed out, choosing optimal $\lambda$ is the practical protocol. However, because this work treats the symmetric Gaussian clusters (i.e., symmetric mean, same rotation invariant covariance) the optimal $\lambda$ becomes infinitely large (at least in the investigated range), where the deviation from the experimental observation with finite $N$ can be significant. Hence, we have shown the results for fixed but various $\lambda$s. If we consider non-symmetric Gaussian clusters, the optimal regularization may become non-trivial, but it goes beyond the scope of the present work, and here we leave it as a future work. We have added an explanation on this point in pages 12-13.
>
> # Other remarks
> > Formulating Assumption 1 as an assumption and not as a property of the problem seems a bit strange to me. Indeed, it is standard that the macroscopic properties do not fluctuate with $D$ in the LSL.
>
> Thanks for pointing this out. We also believe that this is the standard. However, still we have not yet proven this property, so we leave it as an assumption. Also, since placing it in the main text may be redundant,  we have moved Assumption 1 to the derivation section.
>
> > "estiamted" Equation 29
>
> > “Comparison of the comparison of” in Figure 2 might be a typo ?
>
> > "mathop" is not defined (maybe the author meant "ext" instead ?) just after equation 72
>
> Thanks, fixed.
>
> > Figure 1 : not very readable because of the many plots, especially using the two different sizes $N = 2^{10}$ / $2^{13}$ feel redundant
>
> To improve the readability of the figure, we have moved the panels for $N=2^{10}$ to Appendix B.
>
> > In general, the presentation of the Figures could be improved to be more readable. For instance, the legend may not be required in all subplots for Figure 2 and clusters the figure.
>
> Thank your for suggestion for improving the readability. We have grouped together the legends and adjusted so that each subplot is larger than the last.

---

### Review · Reviewer_5dJq · 2024-06-06

**Summary Of Contributions:**

The authors analysed theoretically different mitigation strategies used for imbalanced data, in particular under-sampling (US), sample re-weighing (SW), and under-bragging (UB). In particular, quantifying UB's performance is the main objective of the paper. The authors used methods from statistical physics to reduce the complexity of the problem and evaluate the asymptotic performance of a shallow neural network trained on a Gaussian mixture model. Furthermore, the authors identified a double descent phenomenon associated with reweighing.

**Audience:**

Yes

**Broader Impact Concerns:**

No concerns.

**Claims And Evidence:**

No

**Requested Changes:**

Please, refer to "Strengths and Weaknesses" for more details.

The analysis should be further extended and the results need a more thorough discussion.
At the moment, the paper mostly reports a relevant technical result, but it doesn't improve our understanding of mitigation methods for class imbalance.

**Strengths And Weaknesses:**

# Strengths
- The authors consider a relevant problem of ML.
- The analysis and the comparison of US, UB and SW is novel and significant.
- Up to section 3 (included) the paper is mostly (see below) well-written and nice to read.
- Equation 32 is clear and easy to interpret.
- The double descent phenomenon associated with reweighing is interesting and new.

# Weeknesses
- The discussion of the results is often superficial and stops without analysing in detail the implications of the equations. For instance, the authors obtained the equations for US with replacement but never used them. Another example, the authors decided to consider only the "standard choice" of re-weighing coefficients and the "standard choice" of US, without exploring other possibilities that the equations offered. In ML practice often there is no "standard choice" and people rely on cross-validation steps. So it wouldn't be unrealistic to compare with the optimal value.
- The comparison with US is trivial. The authors considered uniform under-sampling, which in a high-dimensional model is equivalent to the performance of the same model with less data. Indeed, not surprisingly, in Fig.3 the authors observe that the performance for US is constant as the majority class is increased. Two alternatives to make it more interesting would be: not considering uniform undersampling (which is actually a standard practice), or considering US with replacement.
- The discussion of the results seems to be actually lacking an understanding of the results. The authors limit the discussion to an evaluation of the equations, there is no discussion of why, for instance, UB outperforms SW for large majority classes but the vice-versa occurs for small majority classes.
- In the introduction, the authors comment on some inconsistencies in the literature, for instance they mention Nixon, et al. (2020), however the authors never go back to discuss these issues. It is not clear how the initial motivation justifies the analysis and what the results helped us understand.
- The authors stressed in the introduction that most of the literature considers overparameterised models, which appears to be an important aspect of the problem. However, in their discussion of the results, the overparameterisation aspect is never mentioned.
- The authors don't specify the value of K (the number of realisations used in bagging) and never discuss the effect of changing that coefficient in practice. The only discussion is related to Figure 1 which is actually intended to show the validity of the equations rather than the effect on performance.

## Additional comments
- The F-measure is never defined. This is quite important, the authors should probably first introduce it informally in the introduction (when it appears for the first time) and then formally in the problem setup.
- Figures 2, 3, and 6 are suboptimal. In each figure, all the panels share the same legend that ends up occupying a large portion of the figure. The authors could show a single legend (maybe on top or at the bottom of the figures) increasing the individual panels a making the figures more readable.
- Claim 5 is a repetition of Definition 1 adapted to US. It seems quite unnecessary to be reported in the main text, maybe it would be better to move it to the appendix.
- Figure 7 aims to show the performance ratio of UB and SW. However, the choice of the colours is not helping. The authors should use a divergence colourmap, centring the white (or black) to 1, to distinguish the two regions.
- The authors don't discuss [1-4] that consider Gaussian mixture models using a replica-theory approach, in particular [3,4] consider specifically data imbalance.
- Page 3, after generating the tools for asymptotic analysis, the authors concluded with "and so on". They should be specific indicating what is left out of the list, or just end the sentence before.
- Is equation 5 necessary, can it be written in words?

Refs
- [1] Pesce, L., Krzakala, F., Loureiro, B., & Stephan, L. (2023, July). Are Gaussian data all you need? The extents and limits of universality in high-dimensional generalized linear estimation. In International Conference on Machine Learning (pp. 27680-27708). PMLR.
- [2] Loureiro, B., Sicuro, G., Gerbelot, C., Pacco, A., Krzakala, F., & Zdeborová, L. (2021). Learning gaussian mixtures with generalized linear models: Precise asymptotics in high-dimensions. Advances in Neural Information Processing Systems, 34, 10144-10157.
- [3] Mannelli, S. S., Gerace, F., Rostamzadeh, N., & Saglietti, L. (2022). Unfair geometries: exactly solvable data model with fairness implications. arXiv preprint arXiv:2205.15935.
- [4] Loffredo, E., Pastore, M., Cocco, S., & Monasson, R. (2024). Restoring balance: principled under/oversampling of data for optimal classification. arXiv preprint arXiv:2405.09535.

---

> ### Author Response · Authors · 2024-06-11
> **Response to reviewer 5dJq (1/2)**
>
> Thank you for your constructive comments and criticism. We appreciate your consideration to improve our work. Below are the responses to each comment.
>
> # Weakness
> > The discussion of the results is often superficial and stops without analysing in detail the implications of the equations. For instance, the authors obtained the equations for US with replacement but never used them. Another example, the authors decided to consider only the "standard choice" of re-weighing coefficients and the "standard choice" of US, without exploring other possibilities that the equations offered. In ML practice often there is no "standard choice" and people rely on cross-validation steps. So it wouldn't be unrealistic to compare with the optimal value.
>
> Thank you for raising this point. We admit that the word **standard choice** was too naive to use. Indeed, they should be optimized thorough the generalization metric. We have changed the words to **natural initial guess** or **naive candidate** instead of the **standard**. On the other hand, we believe that the chosen parameters are sufficiently legitimate in our setup. (i) the resampling rate; to make the matters simple, in this paper we have taken the position that errors are treated equally for positive and negative exampled data, and have used specificity instead precision, although we do understand that this metric is not the unique choice. Then, using the resampling rate that makes the bias term zero is a natural choice. To check this, we have added the resampling rate dependence of the $F$-measure in Appendix D. (ii) the regularization parameter; as pointed out, choosing optimal $\lambda$ is the practical protocol. However, because this work treats the symmetric Gaussian clusters (i.e., symmetric mean, same rotation invariant covariance) the optimal $\lambda$ becomes infinitely large (at least in the investigated range), where the deviation from the experimental observation with finite $N$ can be significant. Hence, we have shown the results for fixed but various $\lambda$s. If we consider non-symmetric Gaussian clusters, the optimal regularization may become non-trivia, but it goes beyond the scope of the present work, and here we leave it as a future work. We have added an explanation on this point in pages 12-13.
>
> > The comparison with US is trivial. The authors considered uniform under-sampling, which in a high-dimensional model is equivalent to the performance of the same model with less data. Indeed, not surprisingly, in Fig.3 the authors observe that the performance for US is constant as the majority class is increased. Two alternatives to make it more interesting would be: not considering uniform undersampling (which is actually a standard practice), or considering US with replacement.
>
> Thank you for the suggestion to improve the quality of the paper. As suggested, we have included the results for sampling WITH replacement in Section 3.4. The qualitative behavior was almost equal to the case of sampling without replacement.
>
> > The discussion of the results seems to be actually lacking an understanding of the results. The authors limit the discussion to an evaluation of the equations, there is no discussion of why, for instance, UB outperforms SW for large majority classes but the vice-versa occurs for small majority classes.
>
> Actually, intuitive mechanism for the performance difference between UB and SW was described in (Wallace et al., 2011); when the number of data points is small and the data is linearly separable, the position of the classification plane cannot be uniquely determined without regularization even if the cost function is weighted by a multiplicative factor. On the other hand, this intuitive picture cannot explain how much performance difference would be observed quantitatively until the values are actually evaluated precisely. Showing this difference qualitatively using the derived sharp asymptotics is one goal of this paper. Unfortunately, the bounds evaluation could not be shown this time, but an explanation of above situations is added in the introduction.
>
> > In the introduction, the authors comment on some inconsistencies in the literature, for instance they mention Nixon, et al. (2020), however the authors never go back to discuss these issues. It is not clear how the initial motivation justifies the analysis and what the results helped us understand.
>
> Nixon’s results were introduced to motivate the fact that the effect of resampling in GLM/NNs is not fully understood, and are not directly related to the current setting (their results are about DNNs). On the other hand, their study does not resample data depending on the class sizes. Thus, our results indicated that there may be a performance improvement even in DNNs by taking into account the class imbalance structure. We have added a discussion in Section 5.

---

> > ### Author Response · Authors · 2024-06-11
> > **Response to reviewer 5dJq (2/2)**
> >
> > >  The authors stressed in the introduction that most of the literature considers overparameterised models, which appears to be an important aspect of the problem. However, in their discussion of the results, the overparameterisation aspect is never mentioned.
> >
> > As shown in Figure 4 and 7, significant performance improvements are obtained mainly around $\alpha^+ \lesssim 10^1$. This region is below the interpolation threshold for each resampled data (see Figure 5), where is clearly the overparametrized regime. We have added explanations on this point at the last of subsection 4.1 and 4.2.
> >
> > >  The authors don't specify the value of K (the number of realisations used in bagging) and never discuss the effect of changing that coefficient in practice. The only discussion is related to Figure 1 which is actually intended to show the validity of the equations rather than the effect on performance.
> >
> > We admit that the definition of the UB was vague in the previous manuscript. As you guessed, $K\to+\infty$ is used in Section 4, and finite $K$ case is used only for the sanity checks in Section 3. This point is now explicitly explained at the beginning of Section 4.
> >
> > # Other Comments
> > > The F-measure is never defined. This is quite important, the authors should probably first introduce it informally in the introduction (when it appears for the first time) and then formally in the problem setup.
> >
> > Thanks for pointing out this. We have added an formal definition of $F$-measure in Section 2.
> >
> > > Figures 2, 3, and 6 are suboptimal. In each figure, all the panels share the same legend that ends up occupying a large portion of the figure. The authors could show a single legend (maybe on top or at the bottom of the figures) increasing the individual panels a making the figures more readable.
> >
> > Thank your for suggestion for improving the readability. We have grouped together the legends and adjusted so that each subplot is larger than the last.
> >
> > > Claim 5 is a repetition of Definition 1 adapted to US. It seems quite unnecessary to be reported in the main text, maybe it would be better to move it to the appendix.
> >
> > We have moved the Claim 5 to Appendix C.
> >
> > > Figure 7 aims to show the performance ratio of UB and SW. However, the choice of the colours is not helping. The authors should use a divergence colourmap, centring the white (or black) to 1, to distinguish the two regions.
> >
> > Sorry for the misleading diagram. Actually, in most cases, UB estimator has a higher $F$-measure. The heatmap says that UB is overwhelmingly superior when $\alpha^+$ is small and $\alpha^- - \alpha^+$ is large, and that there is almost no performance difference in other cases. In the previous figure, the upper and lower limits of $\alpha^+$ were too small/large, and the area of almost same color was too wide, so we have adjusted the upper and lower limits so that the variation of the color is easy to read.
> >
> > > The authors don't discuss [1-4] that consider Gaussian mixture models using a replica-theory approach, in particular [3,4] consider specifically data imbalance.
> >
> > > [1] Pesce, L., Krzakala, F., Loureiro, B., & Stephan, L. (2023, July). Are Gaussian data all you need? The extents and limits of universality in high-dimensional generalized linear estimation. In International Conference on Machine Learning (pp. 27680-27708). PMLR. [2] Loureiro, B., Sicuro, G., Gerbelot, C., Pacco, A., Krzakala, F., & Zdeborová, L. (2021). Learning gaussian mixtures with generalized linear models: Precise asymptotics in high-dimensions. Advances in Neural Information Processing Systems, 34, 10144-10157. [3] Mannelli, S. S., Gerace, F., Rostamzadeh, N., & Saglietti, L. (2022). Unfair geometries: exactly solvable data model with fairness implications. arXiv preprint arXiv:2205.15935. [4] Loffredo, E., Pastore, M., Cocco, S., & Monasson, R. (2024). Restoring balance: principled under/oversampling of data for optimal classification. arXiv preprint arXiv:2405.09535."
> >
> > Thank you for raising them. We have included these literature into the related works section.
> >
> > > Page 3, after generating the tools for asymptotic analysis, the authors concluded with "and so on". They should be specific indicating what is left out of the list, or just end the sentence before.
> >
> > The example that was not included in the paper is the **second order stein method** (see the literature below). Nowadays, there seems to be many methods to derive the results equivalent to replica symmetric computation and listing all of the existing methods is beyond the author's ability. On the other hand, **and so on** is indeed an ambiguous statement, so we have removed and so on here to finish the sentence.
> >
> > * Bellec, Pierre C., and Cun-Hui Zhang. "Second-order Stein: SURE for SURE and other applications in high-dimensional inference." The Annals of Statistics 49.4 (2021): 1864-1903.
> >
> > > Is equation 5 necessary, can it be written in words?
> >
> > We have removed that equation and replaced it with the text based explanation.

---

> > > ### Comment · Reviewer_5dJq · 2024-06-11
> > >
> > > Thank you for the rebuttal and the changes in the submission.
> > >
> > > I still have a couple of comments:
> > > - I realised I was not explicit enough in my review, I was actually suggesting to play with the re-weighing coefficients beyond that naive choice of them. From Fig.5 we see that the performance for large majority groups couldn't get any worse. So I was wondering if a different choice of re-weighing coefficients would help.
> > > - I didn't realise the colour in Fig.7 was in log-scale. Then, I don't understand why the scale is starting from 0. In old Fig.6 (now Fig.5) we can see that SW is outperforming UB for large dataset sizes and small difference between label sizes. So the negative part of colorbar actually matters. In my previous comment, I suggested using a divergence colourmap centring the crosspoint to 1 without realising that it was in log-scale, my suggestion is now to centre it to 0 (or you could plot the contour $\log F_{UB}/F_{SW}=0$). This would help distinguish the two regions.

---

> > > > ### Author Response · Authors · 2024-06-14
> > > > **Response to the comments 2**
> > > >
> > > > Thank you for further comments. We apologize for the reversal of the order of Figures 5 and 6 in the previous revision. The correct order has been restored.
> > > >
> > > > > I realised I was not explicit enough in my review, I was actually suggesting to play with the re-weighing coefficients beyond that naive choice of them. From Fig.5 we see that the performance for large majority groups couldn't get any worse. So I was wondering if a different choice of re-weighing coefficients would help.
> > > >
> > > > Thank you for the clarification. We have included the case with optimal weighting coefficients.   We found (at least to the author’s surprise) that the performance of SW with optimized weighting coefficients is almost equivalent to the UB method. This may indicate that UB is somehow similar to weighting. The results and discussion are included in Section 4.1, and the abstract, Section 1 and 5 are revised accordingly.  We have also moved the heatmap plot for naive weighting coefficient case to Appendix D.
> > > >
> > > >
> > > > > I didn't realise the colour in Fig.7 was in log-scale. Then, I don't understand why the scale is starting from 0. In old Fig.6 (now Fig.5) we can see that SW is outperforming UB for large dataset sizes and small difference between label sizes. So the negative part of colorbar actually matters. In my previous comment, I suggested using a divergence colourmap centring the crosspoint to 1 without realising that it was in log-scale, my suggestion is now to centre it to 0 (or you could plot the contour $\log F_{UB}/F_{SW}=0$). This would help distinguish the two regions.
> > > >
> > > > We misunderstood your intent and made changes to clarify the difference in $\alpha^+\ll1, \alpha^--\alpha^+\gg1$ in the last revision. This time, we show it in the diverging plot with colormap centered at $0$ (now in Appendix D). Since the regularization parameter is optimized for SW with the naive weighting coefficients, it slightly outperforms UB for $\lambda, \alpha^--\alpha^+\ll1$.

---

> > > > > ### Comment · Reviewer_5dJq · 2024-06-19
> > > > >
> > > > > Thank you for the updates on the submission. I agree with the authors that the finding is surprising, I wonder if this is an artefact of the simplicity of the model or if it is an actual feature in ML.
> > > > >
> > > > > I still have a couple of comments:
> > > > > 1. This is a minor one, I see that in Fig.6 the dashdotted lines is sometimes above the dotted line. This is particularly clear for the plots at large $\alpha^+$. Of course, this is wrong. So the authors should improve their optimiser for the optimum of the weights.
> > > > > 2. Connecting with the incipit to this reply, I wonder if what we are observing in the paper can be observed in real data or if what we are observing is due to the simplicity that comes from using Gaussian data.

---

> > > > > > ### Author Response · Authors · 2024-06-19
> > > > > > **quick response to the first comment**
> > > > > >
> > > > > > Thank you for confirming the revision.
> > > > > > We would like to make a quick response to the first comment before actually making a revision to confirm the current status.
> > > > > >
> > > > > > > This is a minor one, I see that in Fig.6 the dashdotted lines is sometimes above the dotted line. This is particularly clear for the plots at large $\alpha^+$. Of course, this is wrong. So the authors should improve their optimiser for the optimum of the weights.
> > > > > >
> > > > > > First, we briefly summarize the current situation:
> > > > > > * In the first draft, , the optimization was also done for the regularization parameter $\lambda$ for SW with **naive** weighting coefficient
> > > > > > * Even in the current draft, this is true **only** for the naive case. (In the other cases, $\lambda$ is fixed as 1e-3, 1e-1, 1e0)
> > > > > >
> > > > > >  It is purely for this reason that the dashdot line is above the dotted line in Figure 6 (The same is true for Figure 12 in which SW with *naive* coefficients outperforms UB). So, if we fix $\lambda$ for the case of naive weighting coefficient, SW with **naive** coefficients does not exceed UB or SW with optimal coefficients. We guess this is the reviewer's expected behavior.
> > > > > >
> > > > > > This is indeed a bit weird situation, but is purely due to the historical reason of the revision process.
> > > > > >
> > > > > >
> > > > > > Therefore, the possible revision regarding this point would be the categorized into the following two cases:
> > > > > >
> > > > > > 1. Keep Figures 6 and 12 as they are. Add some explanation on the above point.
> > > > > > 2. Fix $\lambda$ for the naive case as well, modify Figure 6, and remove Figure 12 (since SW will no longer exceed UB)
> > > > > >
> > > > > > We can handle both cases immediately, but would Case 2 be preferable?

---

> > > > > > > ### Author Response · Authors · 2024-06-21
> > > > > > >
> > > > > > > The third version was uploaded in response to your request and question.
> > > > > > >
> > > > > > > > This is a minor one, I see that in Fig.6 the dashdotted lines is sometimes above the dotted line. This is particularly clear for the plots at large
> > > > > > > > . Of course, this is wrong. So the authors should improve their optimiser for the optimum of the weights.
> > > > > > >
> > > > > > > As mentioned in the previous comment, the apparent contradiction between SW with naive coefficients and optimal coefficients occurred because $\lambda$ was also optimized only in naive case. We have now fixed $\lambda$ even in the naive case and accordingly revised Figures 6 and 12. We also note that with $\lambda$ fixed, there are no longer cases where naive weighting outperforms UB (see Figure 12).
> > > > > > >
> > > > > > > > Connecting with the incipit to this reply, I wonder if what we are observing in the paper can be observed in real data or if what we are observing is due to the simplicity that comes from using Gaussian data.
> > > > > > >
> > > > > > > Thank you for your comments on important points. Unfortunately, at this point, we do not have a clear answer about the extent of universality, and proving it is beyond what can be addressed in this discussion period.
> > > > > > >
> > > > > > > However, to discuss universality in non-Gaussian data, we have included in Appendix D the results of a binary classification using the *T-shirt/top* and *Shirt* classes of the Fashion-MNIST data. The experimental results are essentially similar to those observed with Gaussian data in Section 4.2; UB and optimally weighted SW provide nearly equivalent performance. Conversely, the performance using naive weighting coefficients drops rapidly as the degree of imbalance increases.
> > > > > > >
> > > > > > > While we do understand that this is not a complete answer, we believe it shows that the observed results are not an artificial artifact that can only be seen with Gaussian data.

---

### Author Response · Authors · 2024-06-11
**Paper Revision**

We would like to thank all the reviewers for their constructive comments. Based on the suggestions and comments, we have revised the manuscript, with revised or added content are shown in red text.  In addition, content that has simply been moved from one place to another is shown in blue text. These parts will revert to the normal font style in the camera-ready version. We hope that the revised manuscript will be suitable for publication in Transactions on Machine Learning Research.

---

### Decision · Action_Editor_iNgT · 2024-07-15

**Recommendation:** Accept with minor revision

**Comment:**

All reviewers considered the manuscript addressess a relevant problem and considered the contribution interesting. However, some concerns were raised on the lack of clarity in the initial manuscript. During the discussion phase, several recommendations were made, and the manuscript saw a substancial rewritting. All reviewers agree that the revised version has been significantly improved, and recommended for acceptance at TMLR.

As a final remark, Reviewer *5dJq* added: "*The authors present a real data experiment in the appendix that agrees with the theoretical results. As a side note, I would suggest pushing this result into the main text as this is relevant for the reader.*".

Therefore, I am happy to recommend this work for acceptance conditioned on this small change.

**Audience:**

I believe this work is of interest to the theoretical machine learning audience at TMLR, in particular those interested in exact asymptotic results.

**Claims And Evidence:**

This work provides a theoretical analysis of different sub-sampling strategies for imbalanced data using tools from statistical physics. Despite being non-rigorous, the tools are standard, and the theoretical results are well supported by finite size experiments.